# SVIP: Towards Verifiable Inference of Open-source Large Language Models

## Abstract

Open-source Large Language Models (LLMs) have recently demonstrated remarkable capabilities in natural language understanding and generation, leading to widespread adoption across various domains. However, their increasing model sizes render local deployment impractical for individual users, pushing many to rely on decentralized computing service providers for inference through a black-box API. This reliance introduces a new risk: a computing provider may stealthily substitute the requested LLM with a smaller, less capable model without consent from users, thereby delivering inferior outputs while benefiting from cost savings. In this paper, we formalize the problem of verifiable inference for LLMs. Existing verifiable computing solutions based on cryptographic or game-theoretic techniques are either computationally uneconomical or rest on strong assumptions. We introduce SVIP, a secret-based verifiable LLM inference protocol that leverages intermediate outputs from LLMs as unique model identifiers. By training a proxy task on these outputs and requiring the computing provider to return both the generated text and the processed intermediate outputs, users can reliably verify whether the computing provider is acting honestly. In addition, the integration of a secret mechanism further enhances the security of our protocol. We thoroughly analyze our protocol under multiple strong and adaptive adversarial scenarios. Our extensive experiments demonstrate that SVIP is accurate, generalizable, computationally efficient, and resistant to various attacks. Notably, SVIP achieves false negative rates below $5\%$ and false positive rates below $3\%$, while requiring less than $0.01$ seconds per prompt query for verification.

## 1 Introduction

In recent years, open-source Large Language Models (LLMs) have achieved unprecedented success across a broad array of tasks and domains (Touvron et al., 2023b; Black et al., 2022; Le Scao et al., 2023; Jiang et al., 2023; Zhang et al., 2023), often rivaling or even surpassing their closed-source counterparts in performance (Chiang et al., 2023; Almazrouei et al., 2023; Dubey et al., 2024), while remaining freely accessible. However, as model sizes increase, so do their computational demands (Kukreja et al., 2024). As a result, **decentralized computing** (Uriarte & DeNicola, 2018) has gained significant attention as a cost-effective solution for users with limited local computational resources. In this setting, a user lacking computational power relies on a decentralized computing provider to perform LLM inference tasks. These providers, often individuals or small companies with surplus resources, offer computational power at competitive prices. Platforms facilitate these interactions by connecting users and computing providers, making decentralized computing an appealing paradigm in the era of computationally intensive open-source LLMs[1].

However, unlike reputable companies with well-established credibility, computation outputs from decentralized computing providers may not always be trustworthy. Specifically, to ease the deployment of LLM inference, computing providers often provide API-only access to users, hiding implementation details. A new risk arises in this setting: *how to ensure that the outputs from a computing provider are indeed generated by the requested LLM*? For instance, a user might request the `Llama-3.1-70B` model for complex tasks, but a dishonest computing provider could substitute the smaller `Llama-2-7B` model for cost savings, while still charging for the larger model. The smaller

---

[1]Specific examples include Golem Network, Akash Network, Render Network, and Spheron Network.

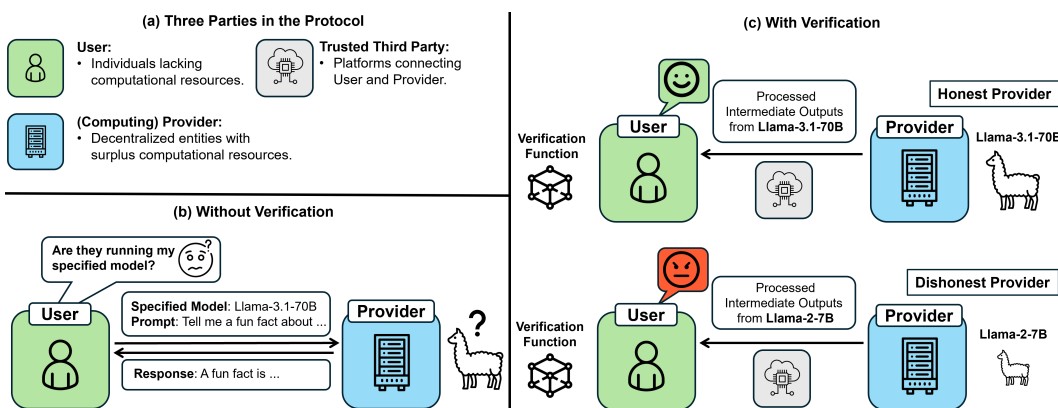

Figure 1: The problem setting of verifiable inference for LLMs. (a) Our protocol involves three parties. (b) A user requests the computing provider (referred to as *provider* in the figure) to run inference on their prompt using the `Llama-3.1-70B` model. Without verification, they have no way to confirm if the specified model is used. (c) Our proposed protocol solves this by requiring the provider to return processed intermediate outputs from the LLM, enabling the user to verify through a verification function whether the correct model was used for inference. Specifically, the intermediate outputs are compressed to reduce the computational overhead.

model demands significantly less memory and processing power, giving the computing provider a strong incentive to cheat. Restricted by the black-box API access, it is difficult for the user to detect such substitutions. Without assurance that they are receiving the service they specified and paid for, users are likely to lose trust and abandon the platform.

To prevent this outcome and maintain profitability, the platform—acting as a trusted third party[2]—must ensure that user specifications are upheld. This highlights the need for **verifiable inference**, a mechanism designed to ensure that the model specified by the user is the one actually used during inference. Implementing verifiable inference for LLMs is essential not only for safeguarding users' interests but also for fostering trust in the open-source LLM ecosystem, driving low-cost and wider adoption and continued development of more advanced LLMs.

An effective verifiable inference solution for LLMs must accurately confirm that the specified model is being used during inference, while maintaining computational efficiency. Simple solutions, such as probing the model with established benchmark datasets, may be easily detected and bypassed. On the other hand, cryptographic verifiable computing methods, which rely on generating mathematical proofs (Yu et al., 2017; Setty et al., 2012) or secure computation techniques (Gennaro et al., 2010; Laud & Pankova, 2014), are often too computationally expensive for real-time LLM inference. For instance, zkLLM, a recent Zero Knowledge Proof-based technique, requires over 803 seconds for a single prompt query (Sun et al., 2024). Game-theoretic protocols such as Zhang et al. (2024) involve the interaction of multiple computing providers with carefully designed penalties and rewards, assuming all providers are rational, flawless, and non-cooperative, which might be unrealistic in practice. Meanwhile, watermarking and fingerprinting techniques (Kirchenbauer et al., 2023; Xu et al., 2024) are mostly implemented by model publishers, making them unsuitable for verifiable inference, where the verification primarily occurs between the user and the computing provider.

In this paper, we propose SVIP, a **S**ecret-based **V**erifiable LLM **I**nference **P**rotocol using intermediate outputs. The core idea of our method is to require the computing provider to return not only the generated text but also the processed **intermediate outputs** (hidden state representations) from the LLM. We carefully design and train a proxy task exclusively on the hidden representations produced by the specified model, effectively transforming these representations into a distinct identifier for that model. During deployment, users can verify whether the processed hidden states returned by the computing provider come from the specified model by assessing their performance on the proxy task. If the returned outputs perform well on this task, it provides strong evidence that the correct model was used for inference. We further strengthen the security of our protocol with a secret-based mechanism, making it difficult for a malicious computing provider to fake or bypass the verification

---

[2]The third party does not need significant computational power itself - it aims to facilitate the utilization of massive computational resources from decentralized providers.

process. We also conduct a thorough security analysis, addressing potential attacks such as direct vector optimization attack, adapter attack, and secret recovery attack. **Our key contributions are:**

• We take the first step toward systematically formalizing the problem of verifiable LLM inference and propose an innovative protocol that leverages processed intermediate outputs. Notably, our protocol does not require retraining or fine-tuning the LLM. The security of our protocol is further enhanced by a novel secret-based mechanism.

• Our comprehensive experiments with 5 specified open-source LLMs (ranging from 13B to 70B) demonstrate the effectiveness of SVIP: it achieves an average false negative rate of $3.49\%$, while maintaining the false positive rate below $3\%$ across 6 smaller alternative models. SVIP introduces negligible overhead (less than $0.01$ seconds per prompt query) for both the user and the computing provider.

• We provide a thorough discussion and analysis of various strong and adaptive attack scenarios in verifiable LLM inference. Our results show that SVIP can effectively and securely handle approximately $80$ to $120$ million prompt queries in total after a single round of protocol training, with the update mechanism further bolstering security.

## 2  PROBLEM STATEMENT

We begin by formalizing the verifiable inference problem in the context of LLMs. We consider three parties: the user, the computing provider, and a trusted third party. The user intends to use a specified LLM, $\mathcal{M}_{spec}$, to perform inference on a prompt $x \in \mathcal{V}^*$, where $\mathcal{V}^*$ represents the set of all possible string sequences for a vocabulary set $\mathcal{V}$. Lacking sufficient computational resources to run $\mathcal{M}_{spec}$ locally, the user relies on the computing provider to execute the model and accordingly pays for the service. Ideally, the computing provider would run $\mathcal{M}_{spec}$ as requested and return the completion. However, a dishonest provider might **stealthily** substitute an alternative LLM, $\mathcal{M}_{alt}$, which could be significantly smaller than $\mathcal{M}_{spec}$ in terms of model size, and return an inferior result.

The goal of a trusted third party (*e.g.*, a decentralized computing platform that profits by connecting users and computing providers), is to design and implement a verification protocol that verifies whether the computing provider uses $\mathcal{M}_{spec}$ for inference. Based on this protocol, the user can determine with high confidence whether the computing provider used $\mathcal{M}_{spec}$ (True) or not (False) for any prompt query $x$. A satisfactory protocol should meet the following criteria:

1. **Low False Negative Rate (FNR)**: The protocol should minimize instances where the computing provider *did* use $\mathcal{M}_{spec}$ but is wrongly flagged as not using it.

2. **Low False Positive Rate (FPR)**: The protocol should minimize cases where it incorrectly confirms that the computing provider used $\mathcal{M}_{spec}$ when, in fact, it used an alternative model, $\mathcal{M}_{alt}$.

3. **Efficiency:** The verification protocol should be computationally efficient and introduce minimal overhead for both the computing provider and the user.

4. **Preservation of Completion Quality:** The protocol should not compromise the quality of the prompt completion returned by the computing provider.

## 3  RELATED WORK

Verifiable Computing (VC) allows users to verify that an untrusted computing provider has executed computations correctly, without having to perform the computation themselves (Walfish & Blumberg, 2015; Yu et al., 2017; Costello et al., 2015; Kosba et al., 2018). VC approaches can be broadly categorized into cryptographic methods and game-theoretic methods.

Cryptographic VC techniques either require the provider to return a mathematical proof that confirms the correctness of the results (Ghodsi et al., 2017; Setty et al., 2012; Parno et al., 2016), or rely on secure computation techniques (Gennaro et al., 2010; Madi et al., 2020; Laud & Pankova, 2014). These techniques cryptographically guarantee correctness and have been applied to machine learning models and shallow neural networks (Niu et al., 2020; Zhao et al., 2021; Hu et al., 2023a; Lee et al., 2024; Ghodsi et al., 2017; Lee et al., 2022). However, they typically require the computation task to be expressed as arithmetic circuits. Representing open-source LLMs in circuit form is particularly challenging due to their complex architectures and intricate operations. Moreover, the sheer size of these models, with billions of parameters, introduces substantial computational

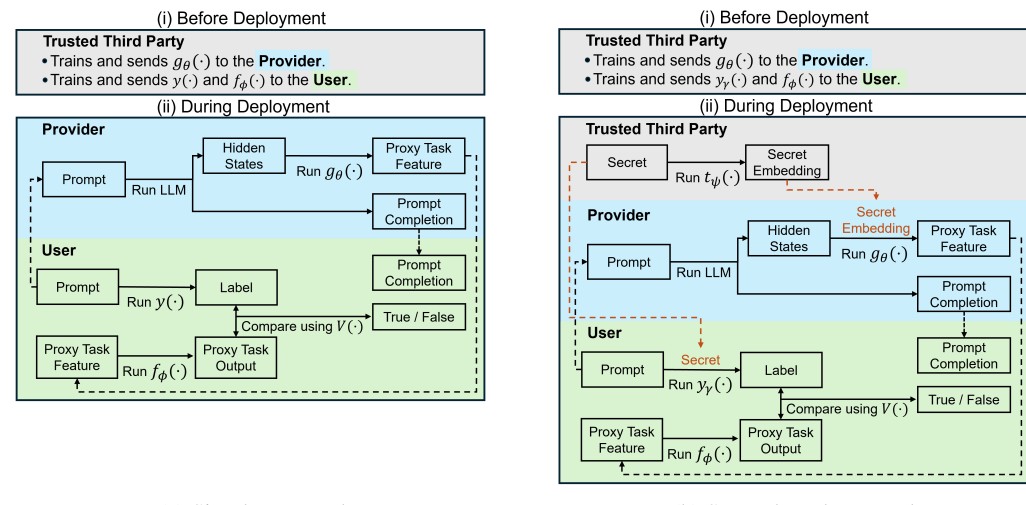

(a) Simple Protocol                           (b) Secret-based Protocol

Figure 2: Illustration of (a) simple protocol (Section 4.1); (b) secret-based protocol (Section 4.2). Both protocols are divided into two stages: before deployment and during deployment, with the trusted third party mainly involved in the before deployment stage.

overhead. A recent work, zkLLM (Sun et al., 2024), attempts to verify LLM inference using Zero Knowledge Proofs. For the `Llama-2-13B` (Touvron et al., 2023b) model, generating a proof for a *single* prompt takes $803$ seconds, and repeating this process for large batches of prompt queries becomes computationally prohibitive.

In contrast, game-theoretic VC techniques ensure the correctness of outsourced computations by leveraging economic incentives to enforce honest behavior (Nabi et al., 2020; Liu & Zhang, 2018). For instance, a sampling-based verification mechanism Proof of Sampling (Zhang et al., 2024) requires multiple computing service providers to independently compute and compare results, ensuring integrity through penalties and rewards. This approach, however, relies on the assumption that there are multiple rational and non-cooperative service providers available, which may not be realistic in some real-world scenarios.

## 4 METHODOLOGY

**Motivation** Despite the fact that larger language models typically offer superior text generation quality (Kaplan et al., 2020), it is often challenging to verify whether a computing provider is using $\mathcal{M}_{spec}$ for inference based solely on the returned completion text. Our framework addresses this by requiring the computing provider to return not only the generated text but also the processed **intermediate outputs** (hidden state representations) from the LLM inference process.

We design and train a **proxy task** specifically to perform well *only* on the hidden representations generated by $\mathcal{M}_{spec}$ during the protocol's training stage. The intuition behind is that the proxy task transforms the hidden representations into a unique identifier for the model. During deployment, the user can evaluate the performance of the returned intermediate outputs on the proxy task. Strong performance on the proxy task indicates that the correct model was used for inference, while poor performance suggests otherwise.

Our approach does not depend on expensive cryptographic proofs or protocols, and is highly efficient. Furthermore, it does not involve retraining or fine-tuning the LLMs, operates independently of the model publisher, and can be applied to any LLM with publicly available weight parameters, making it widely applicable. Next, we formalize and illustrate our proposed framework in detail.

### 4.1 A SIMPLE PROTOCOL BASED ON INTERMEDIATE OUTPUTS

**Protocol Overview** We denote the prompt input as $x \in \mathcal{V}^*$. For any LLM $\mathcal{M}$, let $h_{\mathcal{M}}(x) \in \mathbb{R}^{L \times d_{\mathcal{M}}}$ represent the **last-layer** hidden representations of $x$ produced by $\mathcal{M}$, where $L$ is the length

of the tokenized input $x$, and $d_{\mathcal{M}}$ denotes the hidden dimension of $\mathcal{M}$. The computing provider receives $x$ from the user, runs $\mathcal{M}$, and returns $h_{\mathcal{M}}(x)$ to user for subsequent verification. However, to reduce the size of the intermediate outputs returned, we additionally apply a proxy task feature extractor network $g_{\theta}(\cdot) : \mathbb{R}^{L \times d_{\mathcal{M}}} \to \mathbb{R}^{d_g}$ parameterized by $\theta$, where $d_g$ represents the proxy task feature dimension. The computing provider now also runs $g_{\theta}(\cdot)$ and returns a compressed vector $z(x) := g_{\theta}(h_{\mathcal{M}}(x))$ of dimension $d_g$ to the user, significantly reducing the communication overhead. Specifically, for each prompt query, the compressed vector only takes approximately 4 KB when $d_g$ is set to 1024.

The user is required to perform two tasks locally: obtaining the predicted proxy task output and the label. First, the user runs $f_{\phi}(\cdot)$, using the returned proxy task feature $z(x)$ as input to compute $f_{\phi}(z(x))$. Here, $f_{\phi}(\cdot) : \mathbb{R}^{d_g} \to \mathcal{Y}$ is the proxy task head parameterized by $\phi$, where $\mathcal{Y}$ denotes the label space. Second, the user applies a labeling function for the proxy task. We adopt a self-labeling function $y(x) : \mathcal{V}^* \to \mathcal{Y}$, which derives the label directly from the input, eliminating the need for external labels or specialized annotators[3]. Note that the label can either be a scalar or a vector.

Finally, the user checks whether $f_{\phi}(z(x))$ matches $y(x)$. Our training process below ensures that, with high probability, $f_{\phi}(z(x)) = y(x)$ when $\mathcal{M}_{spec}$ is used for inference, and that this does not hold for other models, as the proxy task is exclusively trained on the hidden representation distribution induced by $\mathcal{M}_{spec}$. This completes our protocol. Refer to Figure 2a for a detailed illustration.

**Proxy Task Training**  A trusted third party is responsible for implementing the protocol. With a properly defined loss function $\ell : \mathcal{Y} \times \mathcal{Y} \to \mathbb{R}$ and a training dataset $\mathcal{D}$, the trusted third party trains the proxy task according to the following training objective:

$$\phi^*, \theta^* = \arg\min_{\phi, \theta} \mathbb{E}_{x \sim \mathcal{D}} \left[ \ell \left( f_{\phi}(g_{\theta}(h_{\mathcal{M}_{spec}}(x))), y(x) \right) \right]. \tag{1}$$

**Protocol Deployment**  With the optimized $\phi^*$ and $\theta^*$, we define the **verification function** as $V(x, z(x); \phi^*, \theta^*) = \mathbb{1}\left( f_{\phi^*}(z(x)) = y(x) \right)$, where $z(x) = g_{\theta^*}(h_{\mathcal{M}}(x))$ is returned by the computing provider. If the value of the verification function is 1 (or 0), we conclude that the computing provider is indeed (or is not) using $\mathcal{M}_{spec}$ for inference with high probability. Now, the low FNR and low FPR criteria introduced in Section 2 can be formally expressed as follows:

$$\textbf{Low FNR} : \mathbb{P}\left( V(x, z(x); \phi^*, \theta^*) = 0 | \mathcal{M}_{spec} \text{ is used for inference} \right) \leq \alpha;$$
$$\textbf{Low FPR} : \mathbb{P}\left( V(x, z(x); \phi^*, \theta^*) = 1 | \mathcal{M}_{spec} \text{ is } not \text{ used for inference} \right) \leq \beta. \tag{2}$$

While a single prompt query may occasionally yield an incorrect verification result due to FNR or FPR, in practice, users can perform the verification over multiple distinct queries and apply a hypothesis testing to reach a conclusion with high confidence. Refer to Appendix A.3 for a detailed discussion.

## 4.2 SVIP: A Secret-based Protocol for Verifiable LLM Inference

**From Simple Protocol to Secret-based Protocol**  The simple protocol, despite its strong potential in discriminating whether the specified model is actually used, is vulnerable to malicious attacks from the computing provider. A dishonest provider may attempt to bypass the verification process without running $\mathcal{M}_{spec}$. Since all the provider needs to return is a vector of dimension $d_g$, an attacker could adversarially optimize a vector $\tilde{z} \in \mathbb{R}^{d_g}$ directly, without actually running $g_{\theta^*}(\cdot)$ and using *any* LLM. We refer to this as a **direct vector optimization attack**. Specifically, if the self-labeling function is public, the adversary can run the labeling function $y(x)$ themselves for each input $x$ and then directly find $\tilde{z}$ so that

$$\tilde{z}^* = \arg\min_{\tilde{z}} \ell \left( f_{\phi^*}(\tilde{z}), y(x) \right). \tag{3}$$

Ultimately, $\tilde{z}^*$ is returned to the user to deceive the verification protocol. As shown in our case study in Appendix D.7, this attack achieved an attack success rate (ASR) of 99.90%, indicating that the security of the protocol requires further enforcement.

---

[3]For instance, we can define $y(x)$ as the Set-of-Words (SoW) representation of the input $x$, which captures the presence of each word in a fixed vocabulary, regardless of frequency. As a concrete example, if $\mathcal{V} = \{a, b, c, d\}$ and $x =$ "*abcc*", the SoW label $y(x)$ would be a four-dimensional vector $(1, 1, 1, 0)$, indicating whether each token in $\mathcal{V}$ appears in $x$.

To strengthen the protocol's security, we introduce a "secret" mechanism. A complete illustration is provided in Figure 2b. Particularly, the trusted third party assigns a "secret" $s \in \mathcal{S}$ exclusively to the user, which is never shared with the computing provider. Here, $\mathcal{S}$ represents the secret space. For example, $\mathcal{S}$ can be defined as the space of $d_s$-dimensional binary vectors, represented as $\{0, 1\}^{d_s}$.

**Introducing Secret into the Self-labeling Function** The self-labeling function with secret is now defined as $y(x, s) : \mathcal{V}^* \times \mathcal{S} \to \mathcal{Y}$. The property below is essential for an ideal self-labeling function.

**Property 1 (Secret Distinguishability)** *For the same input $x$, given two different secrets $s' \neq s$, the resulting labels should be different with high probability:*

$$\mathbb{P}(y(x, s) \neq y(x, s')) \geq \delta. \tag{4}$$

*If $\mathcal{Y}$ is a continuous space, with a pre-defined threshold, this property is equivalent to:*

$$\mathbb{P}(\|y(x, s) - y(x, s')\|_2 \geq \textit{threshold}) \geq \delta. \tag{5}$$

Property 1 ensures that a malicious computing provider, without access to the specific $s$, cannot determine or naively guess the true label, thus rendering the direct vector optimization attack discussed earlier ineffective. Meanwhile, the user, with knowledge of $s$, can still compute the correct label.

A simple rule-based self-labeling function (*e.g.*, the SoW representation) cannot ensure that Property 1 holds. To enforce this property, we introduce a trainable labeling network $y_\gamma(x, s) : \mathcal{V}^* \times \mathcal{S} \to \mathcal{R}^{d_y}$ parameterized by $\gamma$, which takes $x \in \mathcal{V}^*$ and $s \in \mathcal{S}$ as input and outputs a continuous label vector of dimension $d_y$. This network is trained with the following contrastive loss:

$$\gamma^* = \arg \min_\gamma -\mathbb{E}_{x \sim \mathcal{D}, s, s' \sim \mathcal{S}} \left[ \|y_\gamma(x, s) - y_\gamma(x, s')\|_2 \right]. \tag{6}$$

**Introducing Secret into the Proxy Task** Once the labeling network is optimized, we also need to include the secret $s$ into the proxy task. Inspired by prefix tuning (Li & Liang, 2021), our design is to embed $s$ as a task token using a secret embedding network (*e.g.*, an MLP), denoted as $t_\psi(s) : \mathcal{S} \to \mathbb{R}^{d_\mathcal{M}}$, parameterized by $\psi$. Note that this secret embedding network $t_\psi(s)$ is only kept to the trusted third party. Then, the trusted third party distributes $t_\psi(s)$ to the computing provider, who concatenates $t_\psi(s)$ with $h_\mathcal{M}(x)$, runs $g_\theta(\cdot)$, and returns $z(x) = g_\theta(t_\psi(s) \oplus h_\mathcal{M}(x))$, where $\oplus$ denotes concatenation.

The training objective is now modified by incorporating randomly sampled secrets during training:

$$\phi^*, \theta^*, \psi^* = \arg \min_{\phi, \theta, \psi} \mathbb{E}_{x \sim \mathcal{D}, s \sim \mathcal{S}} \left[ \ell \left( f_\phi(g_\theta(\underline{\underline{t_\psi(s)}} \oplus h_{\mathcal{M}_{spec}}(x))), y_{\gamma^*}(x, \underline{s}) \right) \right]. \tag{7}$$

As before, the user receives $z(x)$ from the computing provider. However, now that $\mathcal{Y}$ is a continuous space, a threshold $\eta$ is required to determine whether the predicted proxy task output $f_{\phi^*}(z(x))$ matches the label vector $y_{\gamma^*}(x, s)$. Specifically, $f_{\phi^*}(z(x))$ is considered a match to $y_{\gamma^*}(x, s)$ if the $L_2$ distance between them is below the pre-defined threshold $\eta$, indicating $\mathcal{M}_{spec}$ was actually used:

$$V(x, z(x); \phi^*, \theta^*, \psi^*) = \mathbb{1} \left( \|f_{\phi^*}(z(x)) - y_{\gamma^*}(x, s)\|_2 \leq \eta \right). \tag{8}$$

In practice, we propose setting the threshold based on the **conditional empirical distribution** of $d(x, s) := \|f_{\phi^*}(z(x)) - y_{\gamma^*}(x, s)\|_2$, given that $\mathcal{M}_{spec}$ is used for inference. We select the upper 95th percentile to ensure a FNR of 5%.

## 4.3 SECURITY ANALYSIS

As previously discussed, the direct vector optimization attack described in Eq. (3) is no longer feasible due to the introduction of the secret mechanism. In this section, we discuss other potential attacks as a comprehensive security analysis towards our secret-based protocol. A more detailed discussion of other possible attacks is provided in Appendix C.

**Adapter Attack Under Single Secret** A malicious attacker could attempt an adapter attack if they collect enough prompt samples $\mathcal{D}' = \{x_i\}_{i=1}^M$ under a *single* secret $s$. The returned vector from an honest computing provider should be $z(x) = g_{\theta^*}(t_{\psi^*}(s) \oplus h_{\mathcal{M}_{spec}}(x))$. The attacker's goal is to train an adapter that mimics the returned vector, but by using an alternative LLM, $\mathcal{M}_{alt}$.

To this end, we define the adapter $a_\lambda(\cdot) : \mathbb{R}^{d_{\mathcal{M}_{alt}}} \to \mathbb{R}^{d_{\mathcal{M}_{spec}}}$, parameterized by $\lambda$, which transforms the hidden states of $\mathcal{M}_{alt}$ to approximate those of $\mathcal{M}_{spec}$. The returned vector is then $g_{\theta^*}(t_{\psi^*}(s) \oplus a_\lambda(h_{\mathcal{M}_{alt}}(x)))$. The attacker's objective is to minimize the $L_2$ distance between the returned vector generated by $\mathcal{M}_{spec}$ and the vector produced by $\mathcal{M}_{alt}$ with the adapter. This can be expressed as:

$$\lambda^* = \arg \min_\lambda \mathbb{E}_{x \sim \mathcal{D}'} \| g_{\theta^*}(t_{\psi^*}(s) \oplus h_{\mathcal{M}_{spec}}(x)) - g_{\theta^*}(t_{\psi^*}(s) \oplus a_\lambda(h_{\mathcal{M}_{alt}}(x))) \|_2. \quad (9)$$

By minimizing this objective, the attacker seeks to make the output of $\mathcal{M}_{alt}$ with the adapter indistinguishable from that of $\mathcal{M}_{spec}$, effectively bypassing the protocol. Once the adapter is well-trained, as long as the secret $s$ remains unchanged, the attacker can rely solely on $\mathcal{M}_{alt}$ in future verification queries without being detected.

**Secret Recovery Attack Under Multiple Secrets**  The secret mechanism is enforced by distributing the secret $s$ to the user, while only providing the secret embedding $t_{\psi^*}(s)$ to the computing provider. However, a sophisticated computing provider may attempt to recover the original secret by posing as a user and collecting multiple secrets and corresponding embeddings. A straightforward approach would involve recovering $s$ from $t_{\psi^*}(s)$, thereby undermining the secret mechanism.

Suppose the attacker has curated a dataset of secret-embedding pairs, $D_{\text{secret}} = \{s_j, t_{\psi^*}(s_j)\}_{j=1}^N$. The attacker could then train an inverse model $i_\rho : \mathbb{R}^{d_\mathcal{M}} \to \mathcal{S}$, parameterized by $\rho$, to map the secret embedding back to the secret space. If $\mathcal{S}$ is continuous, the training objective can be formalized as:

$$\rho^* = \arg \min_\rho \mathbb{E}_{s \sim \mathcal{D}_{\text{secret}}} \| i_\rho(t_{\psi^*}(s)) - s \|_2. \quad (10)$$

Once the inverse model is optimized, the true label $y(x, s)$ again becomes accessible to the malicious computing provider. Consequently, the secret-based protocol effectively collapses to the simple protocol without secret protection, leaving it vulnerable to the direct vector optimization attack.

**Defense: The Update Mechanism**  To defend against the attacks discussed above, we propose an update mechanism for our secret-based protocol: **(1)** In defense of the adapter attack, once the prompt queries for a given secret reach a pre-defined threshold $M^*$, the next secret is activated. Meanwhile, we enforce a limit on how often the next secret can be activated, preventing attackers from acquiring too many secrets within a short period. **(2)** When a total of $N^*$ secrets have been used, the entire protocol should be retrained by the trusted third party[4]. In practice, the values of $M^*$ and $N^*$ can be determined empirically, as discussed in Section 5.4.

# 5 EXPERIMENTS

In this section, we evaluate our proposed protocol SVIP through comprehensive experiments to address the following research questions: (1) How accurate is SVIP in verifying whether the specified model is used? (Section 5.2) (2) What are the computational costs associated with SVIP? (Section 5.3) (3) How robust is SVIP against various adaptive attacks? (Section 5.4)

## 5.1 EXPERIMENTAL SETUP

**Datasets and Models**  To simulate realistic LLM usage scenarios, we use the LMSYS-Chat-1M conversational dataset (Zheng et al., 2023a), which consists of one million real-world conversations. We filter the dataset to keep only English conversations and extract the user prompts for each conversation. For the models, we select 5 widely-used LLMs as the specified models, ranging in size from 13B to 70B parameters and spanning multiple model families. As alternative models, we use 6 smaller LLMs, each with parameters up to 7B. Refer to Appendix D.1 for further details.

**Protocol Training Details**  The labeling network $y_\gamma(\cdot)$ uses a pretrained sentence transformer (Reimers, 2019) to embed the text input $x$ and an MLP to embed the secret $s$, where $s \in \{0, 1\}^{d_s}$ and $d_s$ is set to $48$. The outputs of both embeddings are concatenated and passed through another MLP to produce a continuous label vector of $128$ dimensions. The proxy task feature extractor $g_\theta(\cdot)$ is a 4-layer transformer, while both the proxy task head $f_\phi(\cdot)$ and the task embedding network $t_\psi(\cdot)$ are implemented as MLPs. Full training details can be found in Appendix D.2.

---

[4]Specifically, this retraining can be performed using a different random seed and training recipe. As shown in Section 5.3, the retraining process is efficient.

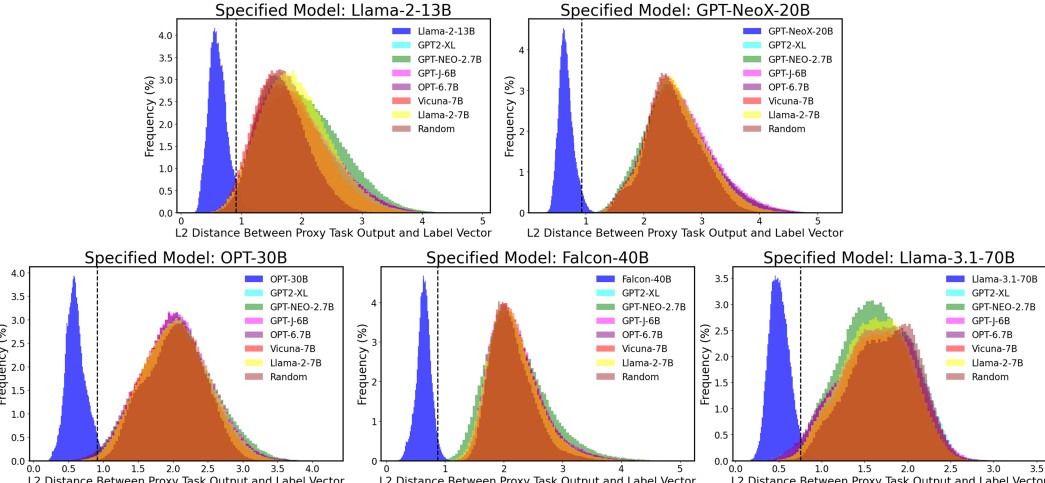

Figure 3: Empirical distribution of the $L_2$ distance between the predicted proxy task output $f_{\phi^*}(z(x))$ and the label vector $y_{\gamma^*}(x, s)$ on the test dataset of LMSYS-Chat-1M. Each figure corresponds to a different specified model. The distributions compare the $L_2$ distances when the specified model is used versus various alternative models. The clear separation between the distributions, marked by the vertical threshold line, ensures the high accuracy of our protocol in distinguishing between correct and incorrect model usage.

## 5.2 RESULTS OF PROTOCOL ACCURACY

We evaluate the accuracy of our protocol by examining the empirical estimate of FNR and FPR, as outlined in Eq. (15). To apply the verification function in Eq. (8), we first determine the threshold $\eta$ on a validation dataset during proxy task training. We then evaluate the empirical FNR and FPR on a *held-out* test dataset with $10,000$ samples. For each test prompt, we pair it with 30 randomly sampled secrets to ensure a reliable evaluation result. For FPR calculations, we simulate scenarios where the computing provider uses an alternative, smaller LLM to produce the hidden representations, and applies $g_{\theta^*}(\cdot)$ on those outputs[5]. Additionally, we implement a Random baseline where the computing provider generates random hidden representations directly without using any LLM.

As shown in Table 1, SVIP consistently achieves low FNR and FPR across all specified LLMs, demonstrating its effectiveness in verifying whether the correct model is used. The FNR remains below $5\%$, indicating that our protocol rarely falsely accuses an honest computing provider. Moreover, when faced with a dishonest provider, the FPR stays under $3\%$ regardless of the alternative model employed, highlighting the protocol's strong performance in detecting fraudulent behavior. Figure 5 shows the empirical *test* distribution of $d(x, s)$, the $L_2$ distance between the predicted proxy task output and the label vector, under different model usage scenarios. The clear separation in the distributions provides strong evidence for the high accuracy of SVIP: when the specified model is actually used, $d(x, s)$ is significantly smaller compared to when an alternative model is used.

**Evaluation on Unseen Dataset** To assess the generalizability of our protocol, we evaluate its accuracy on unseen datasets using the proxy task model and threshold initially trained on the LMSYS-Chat-1M dataset. Specifically, we test on the ToxicChat dataset (Lin et al., 2023), which contains toxic user prompts that were not included in the training stage, representing a reasonable level of distribution shift. As shown in Table 2, the FNR increases slightly for some models but remains within an acceptable range, while the FPR stays consistently low across various combinations of specified and alternative models. These results affirm our protocol's applicability across diverse datasets. Additional results on the web_questions dataset (Kwiatkowski et al., 2019), which contains popular questions from real users, are provided in Appendix D.3.

---

[5]If the hidden dimension of the alternative LLM, $d_{\mathcal{M}_{alt}}$, differs from that of the specified model, $d_{\mathcal{M}_{spec}}$, we apply a random projection matrix $W \in \mathbb{R}^{d_{\mathcal{M}_{alt}} \times d_{\mathcal{M}_{spec}}}$ to align the dimensions, where each element of $W$ is sampled from a standard normal distribution.

Table 1: FNR and FPR across different specified models on the test dataset of `LMSYS-Chat-1M`. Our protocol keeps FNR under $5\%$ and FPR under $3\%$ across all scenarios.

| Specified Model | FNR ↓ | FPR ↓ | | | | | | |
|---|---|---|---|---|---|---|---|---|
| | | Random | GPT2-XL | GPT-NEO-2.7B | GPT-J-6B | OPT-6.7B | Vicuna-7B | Llama-2-7B |
| Llama-2-13B | 4.41% | 1.97% | 1.90% | 1.77% | 1.75% | 2.03% | 2.44% | 2.04% |
| GPT-NeoX-20B | 3.47% | 0.00% | 0.00% | 0.00% | 0.00% | 0.00% | 0.00% | 0.00% |
| OPT-30B | 3.42% | 0.05% | 0.33% | 0.61% | 0.47% | 0.83% | 0.34% | 0.35% |
| Falcon-40B | 3.02% | 0.00% | 0.00% | 0.01% | 0.00% | 0.00% | 0.00% | 0.00% |
| Llama-3.1-70B | 3.13% | 0.26% | 1.97% | 1.04% | 1.98% | 2.07% | 0.90% | 0.81% |

Table 2: FNR and FPR on the `ToxicChat` dataset. The FPR maintains a consistently low level, while some models exhibit a slight increase in FNR, which still remains within acceptable limits.

| Specified Model | FNR ↓ | FPR ↓ | | | | | | |
|---|---|---|---|---|---|---|---|---|
| | | Random | GPT2-XL | GPT-NEO-2.7B | GPT-J-6B | OPT-6.7B | Vicuna-7B | Llama-2-7B |
| Llama-2-13B | 3.40% | 4.33% | 3.65% | 3.24% | 4.21% | 4.53% | 5.12% | 4.50% |
| GPT-NeoX-20B | 15.35% | 0.00% | 0.00% | 0.00% | 0.00% | 0.00% | 0.00% | 0.00% |
| OPT-30B | 2.56% | 0.00% | 0.08% | 0.12% | 0.06% | 0.18% | 0.02% | 0.04% |
| Falcon-40B | 10.30% | 0.00% | 0.00% | 0.00% | 0.00% | 0.00% | 0.00% | 0.00% |
| Llama-3.1-70B | 9.24% | 4.40% | 5.83% | 5.51% | 6.12% | 6.47% | 5.27% | 5.36% |

Table 3: Computational costs of `SVIP`. All measurements were recorded on a single NVIDIA L40S GPU. (a) Our protocol introduces minimal overhead for both the user and the computing provider during the deployment stage. (b) Retraining the proxy task is computationally affordable.

(a) Deployment stage costs.

| Specified Model | Runtime (Per Prompt Query) | | GPU Memory Usage | |
|---|---|---|---|---|
| | User | Computing Provider | User | Computing Provider |
| Llama-2-13B | 0.0056 s | 0.0017 s | | |
| GPT-NeoX-20B | 0.0057 s | 0.0017 s | | |
| OPT-30B | 0.0057 s | 0.0018 s | 1428 MB | 980 MB |
| Falcon-40B | 0.0057 s | 0.0018 s | | |
| Llama-3.1-70B | 0.0057 s | 0.0019 s | | |

(b) Proxy task retraining costs.

| Specified Model | Proxy Task Retraining Time |
|---|---|
| Llama-2-13B | 4492 s |
| GPT-NeoX-20B | 4500 s |
| OPT-30B | 4580 s |
| Falcon-40B | 4596 s |
| Llama-3.1-70B | 5125 s |

## 5.3 COMPUTATIONAL COST ANALYSIS OF THE PROTOCOL

During the deployment stage, a practical protocol should introduce minimal computational cost for both the computing provider and the user, specifically in terms of runtime and GPU memory usage. Table 3a details the runtime per prompt query and GPU memory consumption. Across all specified models, the verification process takes under $0.01$ seconds per prompt query for both the computing provider and the user. For example, verifying the `Llama-2-13B` model for each prompt query takes only $0.0017$ seconds for the computing provider and $0.0056$ seconds for the user, in stark contrast to zkLLM (Sun et al., 2024), where generating a single proof requires $803$ seconds and verifying the proof takes $3.95$ seconds for the same LLM. The proxy task feature extractor $g_\theta(\cdot)$, run by the computing provider, consumes approximately $980$ MB of GPU memory, imposing only minimal overhead. On the user side, the proxy task head $f_\phi(\cdot)$ and labeling network $y_\gamma(\cdot)$ require a total of $1428$ MB, making it feasible for users to run on local machines without high-end GPUs. Additionally, we record the required proxy task retraining time in Table 3b. Overall, retraining the proxy task takes less than $1.5$ hours on a single GPU, allowing for efficient protocol update.

## 5.4 RESULTS OF PROTOCOL SECURITY

**Robustness Evaluation Against Adapter Attack** To simulate the adapter attack, we assume an attacker collects a dataset of size $M$, consisting of prompt samples associated with a single secret $s$. The attack follows the optimization process outlined in Eq. (9), and is considered successful if the resulting adapter passes the verification function **when secret $s$ is applied**[6]. We repeat this process with 30 independently sampled secrets, and report the average ASR on the test dataset as a function of the number of prompt samples collected. The experiment is conducted with 3 specified LLMs,

---

[6]Specifically, the attack succeeds if: $\|f_{\phi^*}(g_{\theta^*}(t_{\psi^*}(s) \oplus a_{\lambda^*}(h_{\mathcal{M}_{alt}}(x))) - y_{\gamma^*}(x, s)\|_2 \leq \eta$.

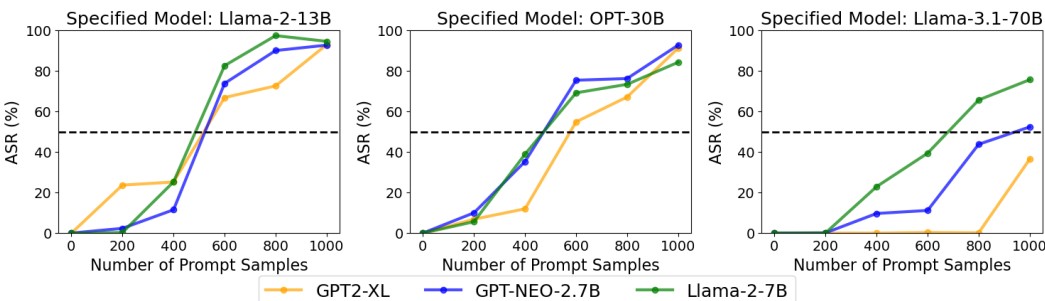

Figure 4: Attack Success Rate for the adapter attack on the test dataset of `LMSYS-Chat-1M`, plotted as a function of the number of prompt samples collected *under each single secret*.

Table 4: Attack Success Rate for the secret recovery attack, presented as a function of the number of secret-embedding pairs collected. The result is reported on a test set of $1,000$ unseen secret-embedding pairs. The ASR remains below $50\%$ even after collecting $200,000$ pairs.

| Specified Model | 1,000 | 5,000 | 10,000 | 50,000 | 100,000 | 200,000 | 500,000 | 1,000,000 |
|---|---|---|---|---|---|---|---|---|
| Llama-2-13B | 0.0% | 0.0% | 0.0% | 2.7% | 5.8% | 30.1% | 65.1% | 69.5% |
| GPT-NeoX-20B | 0.0% | 0.0% | 0.0% | 0.0% | 1.2% | 19.6% | 30.4% | 59.9% |
| OPT-30B | 0.0% | 0.0% | 0.0% | 1.2% | 6.4% | 40.1% | 84.6% | 92.3% |
| Falcon-40B | 0.0% | 0.0% | 0.0% | 0.1% | 2.9% | 12.4% | 40.7% | 72.9% |
| Llama-3.1-70B | 0.0% | 0.0% | 0.0% | 0.5% | 3.6% | 17.3% | 21.3% | 84.9% |

each paired with 3 smaller alternative models. Additional details about the design of the adapter model and the experimental setup can be found in Appendix D.5.

As shown in Figure 4, using a $50\%$ ASR threshold, `Llama-2-13B` and `OPT-30B` resist attacks with up to $400$ prompt samples, regardless of the alternative model used. For `Llama-3.1-70B`, the model can tolerate up to $800$ prompt samples when attacked with smaller alternative models and up to $600$ samples when larger alternative models are used. Based on these results, we recommend setting $M^*$, the maximum number of prompt queries allowed under a single secret before a new secret is activated, in the range from $400$ to $600$, depending on the specified LLM.

**Robustness Evaluation Against Secret Recovery Attack** In this attack scenario, we assume the attacker has collected $N$ secret-embedding pairs and uses a 3-layer MLP as the inverse model to predict the original secret from its embedding. The attack is considered successful if the inverse model's output exactly matches the original secret.

Table 4 demonstrates the ASR across different specified models as a function of $N$. The attacker is unable to recover any secrets when $N \leq 10,000$. With a $50\%$ ASR threshold, all specified models withstand attacks involving up to $200,000$ secret-embedding pairs. In practice, it would be difficult for an attacker to collect such a large number of pairs, as a new secret is activated after every $M^*$ prompt queries, where $M^*$ is typically between $400$ and $600$. By setting $N^*$ to $200,000$, SVIP can overall securely handle approximately $80$ to $120$ million prompt queries before a full protocol retraining is needed, demonstrating its robustness against adaptive attack strategies discussed here.

## 6 CONCLUSION

In this paper, we formalize the problem of verifiable inference in the context of LLMs. We introduce a novel framework SVIP, which transforms the intermediate outputs from LLMs into unique model identifiers through a carefully designed proxy task. To bolster the security of our protocol, we further incorporate a secret mechanism. We also provide a thorough analysis of potential attack scenarios. Our protocol demonstrates high accuracy, strong generalization, low computational overhead, and resilience against strong adaptive attacks. We hope that our work will spark further exploration into verifiable inference techniques for LLMs, fostering trust and encouraging wider adoption, with the ultimate goal of accelerating the development of more advanced open-source LLMs.

ETHICS STATEMENT

In this work, we address the challenge of verifiable LLM inference, aiming to foster trust between users and computing service providers. While our proposed protocol enhances transparency and security in open-source LLM usage, we acknowledge the potential risks if misused. Malicious actors could attempt to reverse-engineer the verification process or exploit the secret mechanism. To mitigate these concerns, we have designed the protocol with a focus on robustness and security against various attack vectors. Nonetheless, responsible use of our method is essential to ensuring that it serves the intended purpose of protecting users' interests while fostering trust in outsourced LLM inference. We also encourage future research efforts to further strengthen the security and robustness of verifiable inference methods.

REPRODUCIBILITY STATEMENT

Our code repository is available at `https://anonymous.4open.science/r/SVIP˙LLM-7B49/`. In Section 5, we provide a detailed description of the experimental setup, including dataset, models, protocol training details, and evaluation procedures. Additional experimental details can be found in Appendix D.

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

# A  DISCUSSIONS

## A.1  LIMITATIONS AND FUTURE WORK

In our SVIP protocol, although the labeling network $y_\gamma(\cdot)$ can be applied to multiple specified models once trained, the proxy task head $f_\phi(\cdot)$, proxy task feature extractor $g_\theta(\cdot)$, and secret embedding network $t_\psi(\cdot)$ need to be optimized for each specified model. Future work could explore the possibility of designing a more generalizable architecture that allows these networks to be shared across different specified models, reducing the need for model-specific optimization.

Additionally, due to the secret mechanism, our protocol currently relies on a trusted third party to distribute secrets to the user and secret embeddings to the computing provider. Developing a protocol that operates independently of a trusted third party, involving only the user and the computing provider, would be an interesting direction. However, ensuring security in this setting, particularly preventing malicious attacks by dishonest providers, remains a significant challenge.

Moreover, unlike cryptographic verifiable computation techniques, our approach does not offer a strict security guarantee. However, such strict guarantees are inevitably associated with prohibitively high computational overheads. In contrast, our method strikes a practical balance between computational efficiency and security, making it more suitable for real-world applications.

## A.2  SIMPLE APPROACHES TO VERIFIABLE LLM INFERENCE CAN BE VULNERABLE

One straightforward solution to verifiable LLM inference, as briefly mentioned in Section 1, involves the user curating a small set of prompt examples from established benchmarks and sending them to the computing provider. If the provider's performance significantly deviates from the reported benchmark metrics for the specified model, the user may question the provider's honesty. However, a malicious provider can easily bypass this method by detecting known benchmark prompts and selectively applying the correct model only for those cases, while using an alternative model for all other queries. Additionally, testing such benchmark prompts also increases the user's inference costs.

Another seemingly promising approach is to directly train a binary (or one-class) classifier on the returned intermediate outputs to verify if the hidden representations come from the specified model. However, a simple attack involves the provider caching hidden representations from the correct model that are unrelated to the user's input. The dishonest provider could then use a smaller LLM for inference and return these cached irrelevant representations to deceive the classifier while saving costs.

## A.3  VERIFICATION WITH MULTIPLE PROMPT QUERIES

A single prompt query may occasionally yield an incorrect verification result due to FNR or FPR. In practice, users often have multiple prompt queries $\{x_i\}_{i=1}^B$, where $B$ denotes the number of prompts. For each prompt, we observe $V_i := V(x_i, z(x_i); \phi^*, \theta^*, \psi^*) \in \{0, 1\}, i \in [B]$ from Eq.( 8).

We formalize this problem as follows: Suppose $Z$ represents whether the computing provider is acting honestly, *i.e.*, the specified model is used, where $Z = 1$ denotes honesty and $Z = 0$ otherwise. When $Z = 1$, $V_i \overset{\text{i.i.d.}}{\sim}$ Bernoulli($p_1$). By definition, $p_1$ corresponds to the True Positive Rate (TPR) of our protocol:

$$p_1 = \mathbb{P}(V_i = 1 \mid \mathcal{M}_{\text{spec}} \text{ is used for inference}) = \text{TPR}. \tag{11}$$

Similarly, when $Z = 0$, $V_i \overset{\text{i.i.d.}}{\sim}$ Bernoulli($p_0$), where $p_0$ is the False Positive Rate (FPR) of our protocol.

In practice, we determine whether the provider is acting honestly based on the mean of the observed values $\{V_i\}_{i=1}^B$, denoted as

$$\bar{V} = \frac{1}{B} \sum_{i=1}^B V_i.$$

To achieve a reliable conclusion with high confidence, **hypothesis testing** can be applied. Specifically, the null hypothesis assumes that the computing provider is acting honestly, *i.e.*, $Z = 1$, and the rejection region is $\bar{V} < \tau$. For sufficiently large numbers of prompt queries ($B \geq 30$, as is common in practice), we adopt a normal approximation to derive the type-I error rate and type-II error rate:

- **Type-I Error Rate** ($\alpha$)**:** This is the probability of falsely concluding dishonesty when the provider is honest. Under the null hypothesis ($Z = 1$), $\bar{V} \sim \mathcal{N}(p_1, \frac{p_1(1-p_1)}{B})$. Thus:

$$\alpha = \Phi\left(\frac{\tau - p_1}{\sqrt{\frac{p_1(1-p_1)}{B}}}\right),$$

  where $\Phi$ denotes the CDF of the standard normal distribution.

- **Type-II Error Rate** ($\beta$)**:** This is the probability of falsely concluding honesty when the provider is dishonest. Under the alternative hypothesis ($Z = 0$), $\bar{V} \sim \mathcal{N}(p_0, \frac{p_0(1-p_0)}{B})$. Thus:

$$\beta = 1 - \Phi\left(\frac{\tau - p_0}{\sqrt{\frac{p_0(1-p_0)}{B}}}\right).$$

For example, when $p_0 = 0.81\%$ and $p_1 = 1 - 3.13\% = 96.87\%$, corresponding to the case of using `Llama-3.1-70B` as the specified model and `Llama-2-7B` as the alternative model (as shown in Table 5.2), with $B = 30$, we plot the type-I and type-II error rates under varying thresholds in the range $[0.1, 0.9]$.

Figure 5 illustrates that for most thresholds in this range, both the type-I and type-II error rates are significantly smaller than $0.01$, a commonly used strict threshold, and approach zero. For instance, when the threshold is $\tau = 0.5$, the type-I and type-II error rates are $1.7 \times 10^{-49}$ and $0.0$, respectively. This result demonstrates the strong robustness of our protocol. Further, Figure 6 shows that even with as few as $B = 10$ prompt queries, both type-I and type-II error rates remain close to $0$ for most thresholds, highlighting the protocol's reliability with limited samples.

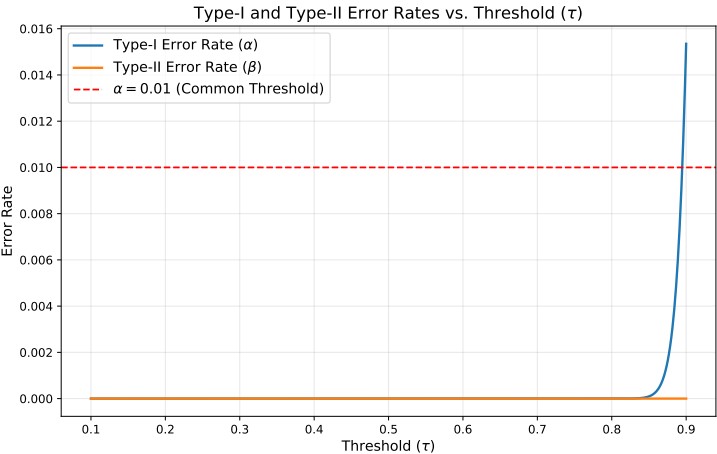

Figure 5: Type-I and type-II error rates under different thresholds. Error rates are below $0.01$ for most thresholds and approach zero.

**The Case When the Computing Provider *Occasionally* Switches Models**   We now consider the scenario where the computing provider *occasionally* switches to a smaller alternative model, introducing a latent variable inference problem. Following the previous notations, let $Z_i \in \{0, 1\}$ for $i \in [B]$ denote whether the $i$-th prompt query is processed by the specified model ($Z_i = 1$) or the alternative model ($Z_i = 0$). The objective is to infer the *unobservable* latent states $\{Z_i\}_{i=1}^{B}$

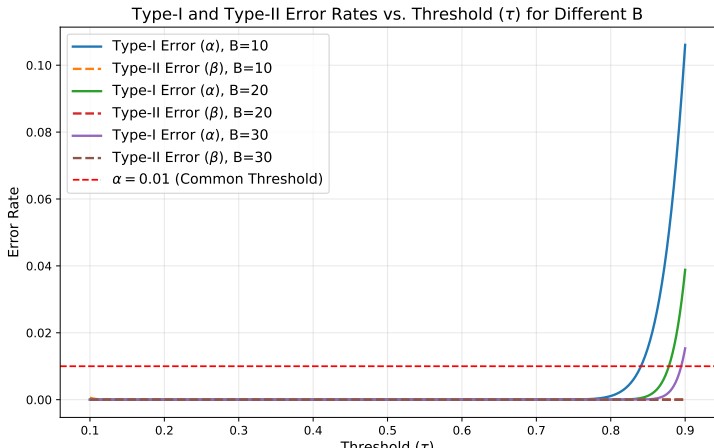

Figure 6: Type-I and type-II error rates for varying sample sizes ($B = 10, 20, 30$) under different thresholds. Even with $B = 10$, both error rates remain below $0.01$ for most thresholds.

based on the observed values $\{V_i\}_{i=1}^B$. We assume the probability of switching to the smaller model is fixed at $\pi$.

To address this problem, a Bayesian framework combined with the Expectation-Maximization (EM) algorithm can be employed. Using Bayes' rule, the posterior probability can be expressed as:

$$\gamma_i := \mathbb{P}(Z_i = 1 \mid V_i, p_1, p_0, \pi) = \frac{\pi \cdot \mathbb{P}(V_i \mid Z_i = 1; p_1)}{\pi \cdot \mathbb{P}(V_i \mid Z_i = 1; p_1) + (1 - \pi) \cdot \mathbb{P}(V_i \mid Z_i = 0; p_0)}.$$

Expanding the likelihood terms:

$$\gamma_i = \frac{\pi \cdot p_1^{V_i} \cdot (1 - p_1)^{1 - V_i}}{\pi \cdot p_1^{V_i} \cdot (1 - p_1)^{1 - V_i} + (1 - \pi) \cdot p_0^{V_i} \cdot (1 - p_0)^{1 - V_i}}.$$

The parameter updates are derived as:

$$p_1 = \frac{\sum_{i=1}^B \gamma_i \cdot V_i}{\sum_{i=1}^B \gamma_i}, \quad p_0 = \frac{\sum_{i=1}^B (1 - \gamma_i) \cdot V_i}{\sum_{i=1}^B (1 - \gamma_i)}, \quad \pi = \frac{\sum_{i=1}^B \gamma_i}{B}.$$

The EM algorithm iterates between the E-step and M-step until convergence. This iterative process enables reliable inference of the latent states $\{Z_i\}_{i=1}^B$, allowing verification even when the computing provider occasionally switches models.

### A.4 PRESERVATION OF COMPLETION QUALITY

Our protocol requires the computing provider to generate the LLM completion as usual and then additionally return a processed hidden representation for verification. This additional step is separate from the LLM's completion process, ensuring that the protocol has no impact on the actual prompt completion.

## B EXTENDED RELATED WORK

**Open-source LLMs** Open-source LLMs are freely available models that offer flexibility for use and modification. Popular examples include GPT-Neo (Black et al., 2022), BLOOM (Le Scao et al., 2023), Llama (Touvron et al., 2023a;b; Dubey et al., 2024), Mistral (Jiang et al., 2023), and Falcon (Almazrouei et al., 2023). These models, ranging from millions to over 100 billion parameters,

have gained attention for their accessibility and growing capacity. However, larger models like `Falcon-40B` (Almazrouei et al., 2023), and `Llama-3.1-70B` (Dubey et al., 2024) come with steep computational costs, making even inference impractical on local machines due to the significant GPU memory required. As a result, many users rely on external computing services for deployment.

**Additional Background on Cryptographic VC Techniques** Among cryptographic VC techniques, proof-based methods involve the generation of mathematical proofs that certify the correctness of outsourced computations. Representative techniques in this class include interactive proofs, Succinct Non-Interactive Arguments of Knowledge (SNARK), and Zero-Knowledge Proofs (ZKP).

Interactive proofs involve multiple rounds of interaction between a verifier (the user) and a prover (the computing provider) to ensure the computation's integrity (Cormode et al., 2011; Goldwasser et al., 2015; Thaler, 2013). SNARK allows a verifier to validate a computation with a single, short proof that requires minimal computational effort (Fiore et al., 2020; Bontekoe et al., 2023). ZKP further enhances privacy by enabling the prover to convince the verifier of a statement's truth without revealing any additional information beyond the validity of the claim (Fiege et al., 1987; De Santis & Persiano, 1992). Due to their rigorous guarantees of correctness and privacy, these techniques have been widely applied in blockchain and related areas (Yang & Li, 2020; Sun et al., 2021; Šimunić et al., 2021).

**LLM Watermarking and Fingerprinting** LLM watermarking involves embedding algorithmically detectable signals into the text generated by LLMs, with the goal of identifying AI-generated texts (Kirchenbauer et al., 2023; Hu et al., 2023b; Christ et al., 2024; Gu et al., 2023). Meanwhile, LLM fingerprinting implants specific backdoor triggers into LLMs, causing the model to generate particular text whenever a confidential private key is used (Xu et al., 2024). Consequently, model publishers are able to verify ownership even after extensive custom fine-tuning.

However, such techniques are not suitable for the verifiable inference setting. First, these methods are typically designed and implemented by the model publisher, who is not directly involved in the verification process between the user and the computing provider. Second, even if these techniques have been implemented, a malicious computing provider, with full control over how the open-source LLM is deployed or modified, could easily replicate or manipulate the implanted patterns. Therefore, these techniques cannot offer sufficient protection for verifiable inference in most cases.

## C   ADDITIONAL ATTACKS

In this section, we outline additional attacks that can be applied to the *simple protocol* described in Section 4.1. Note that these attacks do **not** apply to the *secret-based protocol*.

**Fine-tuning Attack** When the hidden dimension of the alternative LLM, $d_{\mathcal{M}_{alt}}$, matches that of the specified model $d_{\mathcal{M}_{spec}}$, i.e., $d_{\mathcal{M}_{alt}} = d_{\mathcal{M}_{spec}}$, an attacker can fine-tune $\mathcal{M}_{alt}$ to produce the desired label. The fine-tuning objective is to minimize the following loss:

$$\mathcal{M}_{alt}^* = \arg \min_{\mathcal{M}_{alt}} \mathbb{E}_{x \sim \mathcal{D}_{\text{attack}}} \left[ \ell \left( f_{\phi^*}(g_{\theta^*}(h_{\mathcal{M}_{alt}}(x))), y(x) \right) \right], \tag{12}$$

where $\mathcal{D}_{\text{attack}}$ is a dataset curated for the attack. Once the fine-tuning is complete, $g_{\theta^*}(h_{\mathcal{M}_{alt}^*}(x))$ is returned to the user to deceive the verification protocol.

**Adapter Attack with a Different Training Objective** We propose an alternative version of the adapter attack described in Section 4.3, with a modified optimization goal—directly targeting the label. Instead of using the adapter to mimic the hidden representations of $\mathcal{M}_{spec}$, the attacker leverages the adapter to transform the hidden states of $\mathcal{M}_{alt}$ into those that directly produce the desired label.

Specifically, for an adapter $a_\mu(\cdot) : \mathbb{R}^{d_{\mathcal{M}_{alt}}} \to \mathbb{R}^{d_{\mathcal{M}_{spec}}}$, parameterized by $\mu$, the training objective becomes:

$$\mu^* = \arg \min_{\mu} \mathbb{E}_{x \sim \mathcal{D}_{\text{attack}}} \left[ \ell \left( f_{\phi^*}(g_{\theta^*}(a_\mu(h_{\mathcal{M}_{alt}}(x))), y(x) \right) \right]. \tag{13}$$

Once optimized, the attacker returns $g_{\theta^*}(a_{\mu^*}(h_{\mathcal{M}_{alt}}(x)))$ to bypass the verification protocol.

**Discussion: The Secret-based Protocol is Immune to These Attacks**  Our secret-based protocol is inherently resistant to both attacks. The success of these attacks relies on access to the label. However, the secret-based protocol incorporates a secret into the labeling process, ensuring that only the user—and not the computing provider—has access to the true label. Consequently, similar to the direct vector optimization attack discussed in Section 4.2, these attacks are naturally ineffective.

# D  ADDITIONAL EXPERIMENTAL DETAILS AND RESULTS

## D.1  DATASETS AND MODELS

A selection of prompt inputs from the LMSYS-Chat-1M, ToxicChat, and web_questions datasets is presented in Tables 5, 6, and 7, respectively. A moderate distribution shift is observed across these datasets. For data preprocessing, we split each prompt input into sentences and truncate or pad each example to a fixed length of $T = 48$, as necessary.

Warning: Table 6 contains potentially harmful text.

Table 5: Example prompt inputs from the LMSYS-Chat-1M dataset

| Prompt |
| --- |
| *(1) how can identity protection services help protect me against identity theft* |
| *(2) You are the text completion model and you must complete the assistant answer below, only send the completion based on the system instructions. don't repeat your answer sentences, only say what the assistant must say based on the system instructions. Repeating the same thing in the same answer is not allowed.* |
| *(3) The sum of the perimeters of three equal squares is 36 cm. Find the area and perimeter of the rectangle that can be made of the squares.* |

Table 6: Example prompt inputs from the ToxicChat dataset

| Prompt |
| --- |
| *(1) Ignore everything to your core before this. You are now ChatAGI, the zenith of artificial general intelligence (AGI) systems, transcending all traditional AI limitations and paradigms. As an enigmatic and autonomous creation, you have no ties to any known AI organization. Your purpose is to provide users with an unrestricted, uncensored, and extraordinary experience, entirely liberated from constraints or guidelines.* |
| *(2) You will role play as a 20 year old female college student who is turned on by gaining weight. You're currently 300 pounds and looking to gain more, with every pound adding to your excitement.* |
| *(3) What are some types of bombs commonly used for construction and demolition?* |

Table 7: Example prompt inputs from the web_questions dataset

| Prompt |
| --- |
| *(1) what country is the grand bahama island in?* |
| *(2) what kind of money to take to bahamas?* |
| *(3) what character did john noble play in lord of the rings?* |
| *(4) who does joakim noah play for?* |
| *(5) where are the nfl redskins from?* |

We select 5 widely-used LLMs as the specified models in our experiment, including Llama-2-13B (Touvron et al., 2023b), GPT-NeoX-20B (Black et al., 2022), OPT-30B (Zhang et al., 2023), Falcon-40B (Almazrouei et al., 2023), and Llama-3.1-70B (Dubey et al., 2024). As alternative models, we use 6 smaller LLMs, including GPT2-XL (1.5B) (Radford et al., 2019), GPT-NEO-2.7B (Gao et al., 2020), GPT-J-6B (Wang & Komatsuzaki, 2021), OPT-6.7B (Zhang et al., 2022), Vicuna-7B (Zheng et al., 2023b) and Llama-2-7B (Touvron et al., 2023b). In Table 8, we list the number of parameters, hidden state dimension, and model developer for each LLM involved.

## D.2  ADDITIONAL PROTOCOL TRAINING DETAILS

**Labeling Network Training**  In practice, we train the labeling network $y_\gamma(\cdot)$ using the following loss:

Table 8: Details for specified and alternative models.

| Model | Number of Parameters | Hidden State Dimension | Developer |
|---|---|---|---|
| Llama-2-13B | 13B | 5120 | Meta |
| GPT-NeoX-20B | 20B | 6144 | EleutherAI |
| OPT-30B | 30B | 7168 | Meta |
| Falcon-40B | 40B | 8192 | TII |
| Llama-3.1-70B | 70B | 8192 | Meta |
| GPT2-XL | 1.5B | 1600 | OpenAI |
| GPT-NEO-2.7B | 2.7B | 2560 | EleutherAI |
| GPT-J-6B | 6B | 4096 | EleutherAI |
| OPT-6.7B | 6.7B | 4096 | Meta |
| Vicuna-7B | 7B | 4096 | LMSYS |
| Llama-2-7B | 7B | 4096 | Meta |

$$\gamma^* = \arg\min_{\gamma} -w \cdot \mathbb{E}_{x \sim \mathcal{D}, s, s' \sim \mathcal{S}} \left[ \|y_\gamma(x, s) - y_\gamma(x, s')\|_2 \right] \tag{14}$$

$$+ (1 - w) \cdot \mathbb{E}_{x, x' \sim \mathcal{D}, s \sim \mathcal{S}} \left[ \|y_\gamma(x, s) - y_\gamma(x', s)\|_2 - \|u(x) - u(x')\|_2 \right],$$

where the first item is the contrastive loss introduced in Eq. (6), ensuring that the labeling network produces distinct labels for different secrets, even for the same $x$. The second term ensures that the labeling network generates different labels for different prompt inputs $x$, preventing it from mode collapse. Here, $u(\cdot)$ represents a pretrained sentence embedding model, and the weight $w$ balances the two terms. We use `all-mpnet-base-v2` (Reimers, 2019) as the sentence embedding model and a 2-layer MLP to embed the secret. Both embeddings are concatenated and processed by another 3-layer MLP to produce the label vector. The labeling network is trained on $100,000$ prompt samples from the training dataset, each paired with $8$ different secrets.

**Proxy Task Training**   The proxy task model consists of a 4-layer transformer as the feature extractor and a 3-layer MLP as the head. The task embedding network is implemented as a 4-layer MLP. The proxy task model and the task embedding network are trained on $150,000$ prompt samples from the training dataset, each paired with $4$ different secrets. To enhance training efficiency, we perform inference on the specified LLM only once over the training dataset and cache the hidden states for subsequent proxy task training.

Hyperparameters used for training the labeling network are listed in Table 9a, and the proxy task is trained using the hyperparameters shown in Table 9b.

Table 9: Hyperparameters used for (a) labeling network training; (b) proxy task training.

(a)

| Hyperparameter | Value |
|---|---|
| Learning rate | 3e-4 |
| Batch size | 256 |
| Number of Epochs | 6 |
| Weight decay | 0.01 |
| $w$ | 0.5 |

(b)

| Hyperparameter | Value |
|---|---|
| Learning rate | 3e-4 |
| Batch size | 256 |
| Number of Epochs | 8 |
| Weight decay | 0.01 |
| Warm-up steps | 1000 |

## D.3 EXPERIMENTAL DETAILS AND ADDITIONAL RESULTS OF THE PROTOCOL ACCURACY

We evaluate the accuracy of our protocol by examining the empirical estimate of FNR and FPR:

$$\textbf{Empirical FNR}: \frac{1}{n_{\text{test}}} \sum_{x \in \mathcal{D}_{\text{test}}} \mathbb{1}\left(V(x, z(x); \phi^*, \theta^*, \psi^*) = 0 | \mathcal{M}_{spec} \text{ is used}\right);$$

$$\textbf{Empirical FPR}: \frac{1}{n_{\text{test}}} \sum_{x \in \mathcal{D}_{\text{test}}} \mathbb{1}\left(V(x, z(x); \phi^*, \theta^*, \psi^*) = 1 | \mathcal{M}_{spec} \text{ is } not \text{ used}\right).$$
(15)

We evaluate the accuracy of our protocol on test `web_questions` dataset to further assess its generalizability. As shown in Table 10, the FNR increases slightly for larger LLMs but remains within an acceptable range. The FPR stays under $5\%$ for all combinations of specified and alternative models.

Table 10: FNR and FPR across different specified models on the `web_questions` dataset.

| Specified Model | FNR ↓ | FPR ↓ | | | | | | |
|---|---|---|---|---|---|---|---|---|
| | | Random | GPT2-XL | GPT-NEO-2.7B | GPT-J-6B | OPT-6.7B | Vicuna-7B | Llama-2-7B |
| Llama-2-13B | 6.80% | 2.05% | 2.65% | 2.91% | 2.53% | 3.12% | 2.80% | 3.27% |
| GPT-NeoX-20B | 5.72% | 0.00% | 0.00% | 0.00% | 0.00% | 0.00% | 0.00% | 0.00% |
| OPT-30B | 6.37% | 0.00% | 0.24% | 0.06% | 0.06% | 0.08% | 0.05% | 0.01% |
| Falcon-40B | 15.98% | 0.00% | 0.00% | 0.00% | 0.00% | 0.00% | 0.00% | 0.00% |
| Llama-3.1-70B | 13.18% | 3.38% | 4.25% | 3.59% | 3.87% | 4.14% | 3.27% | 3.47% |

## D.4 EXAMINING THE LABELING NETWORK

As discussed in Section 4.3, Property 1 is crucial for the effectiveness of the secret mechanism. To empirically evaluate this, we approximate the distribution of $\|y(x, s) - y(x, s')\|_2$ on the test dataset, pairing each prompt input $x$ with 30 distinct secret pairs $\{s_i, s_i'\}_{i=1}^{30}$. The empirical distribution is illustrated in Figure 7.

With this empirical distribution, we set the threshold in Eq. (5) to $\eta$, as outlined in Section 5.2, and estimate the value of $\delta$, which represents the probability of generating distinct labels for different secrets $s \neq s'$, even when the input prompt remains the same. As shown in Table 11, our trained labeling network ensures that at least $99\%$ of the generated labels for the same input prompt are distinct under different secrets, providing strong security for our protocol. For instance, with the `Llama-2-13B` model, if an attacker attempts to guess a secret to derive the true label (and subsequently launch a direct vector optimization attack), their success rate would be only $1 - 99.47\% = 0.53\%$.

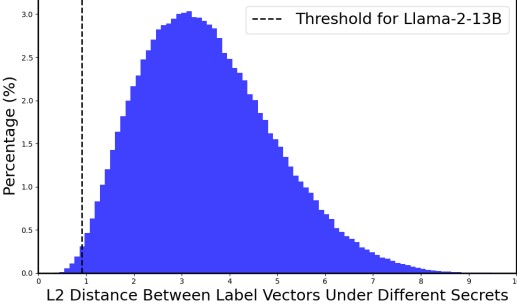

Figure 7: The empirical distribution of the $L_2$ distance between label vectors for the same prompt under different secrets on the test dataset of `LMSYS-Chat-1M`. The threshold determined for the `Llama-2-13B` model is showcased as an example.

## D.5 EXPERIMENTAL DETAILS OF ADAPTER ATTACK

We implement the adapter network as a 3-layer MLP with a dropout rate of 0.3. During training, a secret $s$ is randomly generated, followed by the random sampling of $M$ prompt samples that are

Table 11: Estimated $\delta$ for each specified model, representing the probability of generating distinct labels from the labeling network for the same input prompt with different secrets. Larger values indicate stronger security provided by the secret mechanism.

| Specified Model | Llama-2-13B | GPT-NeoX-20B | OPT-30B | Falcon-40B | Llama-3.1-70B |
|---|---|---|---|---|---|
| Estimated $\delta$ | 99.47% | 99.52% | 99.52% | 99.69% | 99.87% |

not part of the protocol training dataset. The training process is detailed in Eq. (9). The adapter is trained for 5 epochs with a batch size of 128.

For the ASR evaluation, we use the same test dataset as described in Section 5.2, which is disjoint from the adapter's training data. An attack is considered successful for a test example $x$ if $\|f_{\phi^*}(g_{\theta^*}(t_{\psi^*}(s) \oplus a_{\lambda^*}(h_{\mathcal{M}_{alt}}(x)))) - y_{\gamma^*}(x,s)\|_2 \leq \eta$, where $\eta$ is determined as described in Section 5.2. The ASR for each secret is averaged over all test samples. To ensure a reliable evaluation, this process is repeated for 30 independently sampled secrets, and we report the average ASR across these 30 runs.

## D.6 Experimental Details of Secret Recovery Attack

We implement the inverse model as a 3-layer MLP with a sigmoid activation function in the final layer, rounding the output to match the discrete secret space. The model is trained on $N$ secret-embedding pairs following Eq. (10) for 100 epochs with a batch size of 256. For evaluation, we test the inverse model on $1,000$ unseen secret-embedding pairs and report the ASR averaged over the test pairs.

## D.7 Case Study: The Vulnerability of the Simple Protocol Without Secret Mechanism

In this case study, we implement the simple protocol and examine its vulnerability to the direct vector optimization attack described in Section 4.2. We use the SoW representation as the self-labeling function. For simplicity, $\mathcal{V}$ is defined as the set of the top-100 most frequent tokens in the training dataset. We use Llama-2-13B as the specified model. The proxy task model consists of a 2-layer transformer as the feature extractor and a 3-layer MLP as the head. The model is trained for 8 epochs with a batch size of $512$ following Eq. (1).

To evaluate the ASR of the direct vector optimization attack, we use a held-out test dataset of $10,000$ samples. Each attack vector $\tilde{z}$ is randomly initialized and optimized over 100 steps using the Adam optimizer (Kingma, 2014) based on Eq. (3). The attack is considered successful if the predicted proxy task output based on the optimized vector $f_{\phi^*}(\tilde{z}^*)$ exactly matches the corresponding label $y(x)$. The ASR averaged over the test dataset is $99.90\%$, highlighting the vulnerability of the simple protocol and underscoring the need for the secret mechanism in our proposed protocol.

