# OpenReview forum: "SVIP: Towards Verifiable Inference of Open-source Large Language Models"
_ICLR.cc/2025/Conference — Submitted to ICLR 2025_

### Official Review · Reviewer_9Dbq · 2024-11-01

**Soundness:** 3
**Presentation:** 2
**Contribution:** 4
**Rating:** 6
**Confidence:** 4

**Summary:**

The paper proposes Secret-based Verifiable LLM Inference Protocol (SVIP), as a method that allows black-box API users to verify that the LLM the provider claims to offer to them is actually the one that returns the responses. Therefore, the LLM is trained on a proxy task (for the intermediate representations), such that these intermediate responses become a unique identifier of the model. Whether or not the model is the claimed one can then be assessed if the model provider returns intermediate representations along with the outputs.

**Strengths:**

The paper has significant strengths:
- The paper addresses a highly relevant and novel problem of verifying the authenticity of the model behind an API, which is essential for trust in model-based services and has broad implications for the field.
- Reducing the size of intermediate representations to minimize communication overhead is a practical and effective approach, improving the feasibility of real-time verification in resource-constrained environments.
- The paper’s thorough consideration of various attack vectors enhances the robustness of the proposed method and demonstrates an awareness of potential vulnerabilities.
- Experimentation considers multiple LLMs.

**Weaknesses:**

While I really enjoyed reading the paper and found it exciting to see this first step forward towards a new threat vector, I currently see significant weaknesses. If these can be resolved / clarified during the rebuttal process, and if the paper can be updated accordingly, I am willing to raise my score to an accept.

**There is not sufficient protocol setup description**

Even after multiple reads of the paper, I still struggle to see some details of the actual deployment protocol for the method. In particular, I am searching for answers for the following questions:
1. Why are the last layer hidden representations chosen?
2. Where does the user get the proxy-task head from?
3. Where does the user get y(x) from?
4. Where is the actual user's task solved, and how does the actual query need to relate to the "verification query"? Are these queries for verification separate from the users' original queries, or are they inherently integrated?
5. Who trains the trainable labeling network? Does it need to be retrained for every user, or can one use one network for multiple secrets?
What I suggest to include into the paper is a more expressive Figure 1. One that really illustrates the protocol. Who generates what, passes it on to whom etc.
For 1., (4.), and 5., additions to the writing would probably resolve the questions.

**Limitations in terms of description of the experimental setup**

Even after reading the full appendix as well, I do not have a mental model of what concretely:
1. The labeling function is composed of.
2. What the proxy tasks are? Are the some classification task? If so on what dataset? How is that chosen?
3. In a similar vein, it would be great to describe and given an example on how the secrets look like in practice. Are those single words? How are they chosen?

**Limitations in experimental evaluation and description**

Even after the experimental section and the appendix, I am not clear about how a user query really looks like. Additionally, I think it would be sensible to include the following results:
1. The whole presented results focus on the success of the protocol, but not on task accuracy. It would be necessary to show what is the impact of the protocol on the task that the user actually wants to solve.
2. The compute time in Table 3 only focuses on the verification. But what are the computation costs for the trusted party of doing all the overhead? It seem that they are actually the ones with the heavy load.


**Conceptual Limitations**

While I think that it is not necessary to resolve those, I feel like it would be best to discuss the following limitations in the main paper and not to defer them to the appendix where current limitations are located.
- Limitation 1: Works only if the LLM provider is willing to take an open source model, or otherwise share their LLM with the trusted party such that they can embed a task into it. In reality, I do not see much hope that OpenAI would disclose their model for this kind of training to another party.
- Limitation 2: as the work shows, without using a secret, the approach is susceptible to attacks. Hence, it needs the additional user-secret, which requires the presence of the trusted party and significant overhead for them. Who's gonna pay for that?

Minor: The paper states: "while some models exhibit a slight increase in FNR, which still remains within acceptable limits": it is unclear who would define what an acceptable limit is?





**Minor Writing Suggestions**
- Abstract: missing S: "that leverages intermediate outputs from LLM*s*"

**Questions:**

1. Why are the last layer hidden representations chosen?
2. Where does the user get the proxy-task head from?
3. Where does the user get y(x) from?
4. Where is the actual user's task solved, and how does the actual query need to relate to the "verification query"? Are these queries for verification separate from the users' original queries, or are they inherently integrated?
5. Who trains the trainable labeling network? Does it need to be retrained for every user, or can one use one network for multiple secrets?
What I suggest to include into the paper is a more expressive Figure 1. One that really illustrates the protocol. Who generates what, passes it on to whom etc.


6. What the proxy tasks are? Are the some classification task? If so on what dataset? How is that chosen?
7. How do the secrets look like in practice? Are those single words? How are they chosen?

---

> ### Author Response · Authors · 2024-11-21
> **Thank you for your feedback!**
>
> We thank you for recognizing the novelty of our setting and the significant contributions of our work. We greatly appreciate your questions on the details of our paper, and we believe our explanations below have addressed all your questions. Additionally, we have **revised our paper to improve clarity on the technical aspects based on your suggestions**. We hope you can re-evaluate our paper based on our response, and please feel free to let us know if you have any additional questions.

---

> ### Author Response · Authors · 2024-11-21
> **Weakness 1**
>
> **Weakness 1: Why are the last layer hidden representations chosen?**
>
> **Re.:** The last-layer hidden representations are chosen because they capture high-level features that are most indicative of the model's final output, following [1][2].
>
> [1] Devlin et al. (2019). "BERT: Pre-training of Deep Bidirectional Transformers for Language Understanding".
>
> [2] Radford et al. (2018). "Improving Language Understanding by Generative Pre-Training".

---

> ### Author Response · Authors · 2024-11-21
> **Weakness 2**
>
> **Weakness 2: Where does the user get the proxy-task head from?**
>
> **Re.:** The user receives the proxy-task head model, which is trained by the trusted third party, directly from the trusted third party.
>
> We have updated Figure 2 to better illustrate this process.

---

> ### Author Response · Authors · 2024-11-21
> **Weakness 3**
>
> **Weakness 3: Where does the user get y(x) from?**
>
> **Re.:** In general, y(x) (or y(x,s)) is a function distributed by the trusted third party to the user before the deployment of the verification protocol. We have updated Figure 2 to make this more clear.
>
> In the simple protocol (Section 4.1), y(x) is a parameter-free labeling function that is publicly provided by the trusted third party.
>
> In the secret-based protocol (Section 4.2), y(x,s) is a labeling network trained by the trusted third party and given directly to the user.

---

> ### Author Response · Authors · 2024-11-21
> **Weakness 4**
>
> **Weakness 4: Where is the actual user's task solved, and how does the actual query need to relate to the "verification query"? Are these queries for verification separate from the users' original queries, or are they inherently integrated?**
>
> **Re.:** The "verification query" and the "actual query" are the same. Our verification process does not require additional queries. The user's demand is simply a prompt query (e.g., "Generate a summary of this text…"), which the user expects the computing provider to complete using the specified LLM and return.
>
> The purpose of verification is to ensure that the computing provider is indeed using the specified LLM to process the requested prompt. In our protocol, alongside providing the ordinary outputs for the user's task, the computing provider also returns proxy task features, which allows verification of inference results. We have updated Figure 2 to make this more clear.

---

> ### Author Response · Authors · 2024-11-21
> **Weakness 5**
>
> **Weakness 5: Who trains the trainable labeling network? Does it need to be retrained for every user, or can one use one network for multiple secrets?**
>
> **Re.:** The labeling network is trained by the trusted third party. Once trained, it can be applied to various specified LLMs, for any user and any secret, without retraining, as it simply provides a label.
>
> As for the request for a clearer figure, we have updated Figure 2 on page 4 to make the processes clearer, explicitly illustrating who trains, generates, and receives each component in the protocol. We hope this revision resolves any ambiguity and enhances understanding.

---

> ### Author Response · Authors · 2024-11-21
> **Weakness 6**
>
> **Weakness 6: What the labeling function is composed of.**
>
> **Re.:** For the simple protocol (Section 4.1), the labeling function y(x) takes x as input and outputs a vector or scalar. For example, it can be defined as the Set-of-Words (SoW) representation of x, capturing the presence of each word in a fixed vocabulary. This is explained in Footnote 3 on page 5, where we provide a detailed example.
>
> For the secret-based protocol (Section 4.2), the labeling function y(x,s) takes both the input x and a generated secret s, and produces a continuous d_y-dimensional vector as output.
>
> For reference, the simple protocol definition is on Line 228, and the secret-based protocol definition is on Line 276.

---

> ### Author Response · Authors · 2024-11-21
> **Weakness 7**
>
> **Weakness 7: What the proxy tasks are? Are they some classification task? If so on what dataset? How is that chosen?**
>
> **Re.:** The purpose of the proxy task is to transform the hidden representations into a unique identifier for the specified model. This ensures that **the proxy task performs well only when the specified model is actually used for inference**. In that way, if we observe a good performance on the proxy task for some returned hidden representations, we can confidently conclude the computing provider is acting honestly.
>
> In the simple protocol (Section 4.1), the proxy task can be either classification or regression, depending on the labeling function. In the secret-based protocol (Section 4.2), the proxy task is regression, as our labeling network outputs a continuous vector.
>
> The third party trains the proxy task feature extractor and head exclusively on the hidden representations generated by the specified model using an open chat dataset with diverse prompts (e.g., LMSYS-Chat-1M). Furthermore, in Section 5.2 and Appendix D.3, we demonstrate the protocol’s robustness to data distribution shifts with experimental results on datasets including ToxicChat and web questions.

---

> ### Author Response · Authors · 2024-11-21
> **Weakness 8**
>
> **Weakness 8: In a similar vein, it would be great to describe and given an example on how the secrets look like in practice.**
>
> **Re.:** The secret s in our protocol is a 48-bit binary vector, denoted as s $\in \\{0,1\\}^{48}$, as specified in Line 371. We have revised the writing to ensure better clarity.

---

> ### Author Response · Authors · 2024-11-21
> **Weakness 9**
>
> **Weakness 9: It would be necessary to show what is the impact of the protocol on the task that the user actually wants to solve.**
>
> **Re.:** Our protocol requires the computing provider to generate the LLM completion as usual and then **additionally** return a processed hidden representation. Therefore, our protocol has no impact on the actual prompt completion (i.e., the task the user wants to solve) at all.
>
> We have added a discussion in Appendix A.4 to make this point more clear.

---

> ### Author Response · Authors · 2024-11-21
> **Weakness 10**
>
> **Weakness 10: But what are the computation costs for the trusted party of doing all the overhead?**
>
> **Re.:** Even for the largest model (70B), this training requires less than 1.5 hours, demonstrating that the trusted third party’s computational load is quite manageable. Please refer to Table 3(b) on page 9, where we show the proxy task retraining time, which is essentially the training time itself.

---

> ### Author Response · Authors · 2024-11-21
> **Weakness 11**
>
> **Weakness 11: Works only if the LLM provider is willing to take an open source model, or otherwise share their LLM with the trusted party such that they can embed a task into it.**
>
> **Re.:** First, we clarify that there is no "LLM provider" in our setting. This term differs from the "computing provider" we discuss. For instance, an LLM provider could be a large tech company, such as Meta or DeepSeek, while the computing provider is typically an entity or individual with surplus computational resources.
>
> Our work specifically considers cases where the computing provider uses an open-source LLM for inference, as stated clearly in both our title and introduction. Close-sourced models like those from OpenAI are outside the scope of our discussion, since these models are not suitable for the decentralized computing setting studied in our paper, where the inference workloads are distributed into individuals and small companies at a low cost.

---

> ### Author Response · Authors · 2024-11-21
> **Weakness 12**
>
> **Weakness 12: Hence, it needs the additional user-secret, which requires the presence of the trusted party and significant overhead for them. Who's gonna pay for that?**
>
> **Re.:** The third party can be understood as a platform connecting two entities: (1) computing providers with surplus computational capacity to rent out at competitive prices, and (2) users needing computational resources. Roughly speaking, it can be seen as a computing rental marketplace, as seen with platforms such as the Golem Network [1], the Akash Network [2], the Render Network [3], and the Spheron Network [4]. This platform has a strong motivation to monitor computing providers' model usage to maintain user trust; without such oversight, users would likely stop using this platform due to fake or low-quality inference results
>
> Moreover, as we highlighted in our response to Weakness 10, the overhead for the trusted third party is not "significant" given the manageable training time required.
>
> [1] https://www.golem.network/
>
> [2] https://akash.network/
>
> [3] https://rendernetwork.com/
>
> [4] https://www.spheron.network/

---

> ### Author Response · Authors · 2024-11-21
> **Weakness 13**
>
> **Weakness 13: It is unclear who would define what an acceptable limit is?**
>
> **Re.:** In cases without distribution shift, all FPRs and FNRs are below 5%. Even with distribution shift, the highest observed error rate is approximately 15%. As outlined in the general response above, conclusions about the provider’s honesty are not based on a single query but rather on multiple independent queries. By maintaining FPRs and FNRs below 15% for individual prompts, user can draw a **final conclusion** about the provider’s honesty with high confidence.
>
> Intuitively, even if an error occurs on one prompt (with a probability <15%), multiple independent prompt queries make it highly unlikely for errors to persist across the entire batch, leading to a highly reliable conclusion. Empirically, using the hypothesis testing framework detailed in Appendix A.3, when FPR = 15\% and FNR = 15\%, basing the conclusion on 30 prompts with a threshold of  $\tau = 0.5$ results in type-I and type-II error rates of approximately $3.96 × 10^{-8}$ and  $3.96 × 10^{-8}$, respectively.

---

> ### Author Response · Authors · 2024-11-21
> **Weakness 14**
>
> **Weakness 14: Abstract: missing S: "that leverages intermediate outputs from LLMs"**
>
> **Re.:** Thank you for spotting this. We have revised the abstract accordingly.

---

> ### Author Response · Authors · 2024-11-24
> **Thank you for your re-evaluation.**
>
> We are glad our responses and revised version addressed your concerns. Thank you for re-evaluating our paper!

---

### Official Review · Reviewer_jCVz · 2024-11-05

**Soundness:** 2
**Presentation:** 2
**Contribution:** 2
**Rating:** 3
**Confidence:** 3

**Summary:**

This paper proposes a protocol to verify that a particular model is being used. Instead of using verifiable computing or watermarking, this work proposes to use a proxy task to discern between the last-layer representations of the chosen model and other models. Thus, by releasing a projection of these representations, a user can now detect if the desired model was actually used. This work includes investigation of adaptive attacks showing that the final protocol is robust to these attacks and empirical evaluation showing high success at distinguishing models.

**Strengths:**

- There are many models (llms) tested.

- The protocol is shown to be effective at distinguishing between two models.

- Some adaptive attacks are considered and the modified protocol is robust to these.

- The protocol is highly efficient, both in terms of compute and communication overhead.

**Weaknesses:**

- Lacking motivation, see questions 1-3. It is unclear to me the impact of this problem setting/solution. It seems that model providers are already heavily disincentivized from “swapping models”, could be caught (and if not, it then wouldn’t be a problem), or may already be on a path to removing the need to “swap models”. I welcome a response to these questions.

- The protocol is unclear. For example, how are the inputs to the proxy task generated? Does the choice in the labeling function matter?

- Figure 2 seems incorrect. Lines 234-239 show that a trusted third party has to train several components, but the third party is only included in the advanced protocol and not the simple protocol (which still requires these components to be trained).

- Much of the security analysis assumes a highly malicious adversary. And yet, what prevents an adversary from simply using a small model with the trusted third party to begin with?

- Limited security guarantees. Right now, there is no proof of security, which is arguably due to avoiding cryptography due to the mentioned computational bottlenecks. However, this protocol does not appear to be robust. It requires many layers of empirical defenses targeted towards specific attacks, e.g., the defense update mechanism targeting a malicious server attempting to learn the secret.

- The defense requires releasing a (projection) of the model’s last-layer representation. This can make the model susceptible to model-stealing attacks. Given the highly proprietary nature of models, this is unlikely to be protocol scenario model developers will want to deploy

- Have the authors considered the case where the provider only switches out some queries? It is unlikely that the model provider will swap all queries.

**Questions:**

- Many model providers are already training cheaper smaller models and showing how these models achieve near-par performance to larger models. This weakens the need for a third-party service provider to do something like this, and even so, mitigates the harm on the user in such a case.

- Right now, model size is far from the best way to choose a model. Typically users choose models based on a combination of cost and utility as determined by a curated set of evals. Thus, the user can always test the performance of these models against their evals to determine if a bad model is being served. Thus, detection seems easy. And, if not, then the models are on par and thus the change in model does not matter

- The damages of being caught seem to far outweigh the benefits.

---

> ### Author Response · Authors · 2024-11-21
> **Thank you for your feedback! We believe there are key misunderstandings in the review.**
>
> Thank you for your thoughtful feedback. We believe there are **key misunderstandings** about our setting. Before addressing each question below, we would like to clarify our setting.
>
> Our setting involves a user and a computing provider (not the “model provider” as the reviewer suggests). The user requests the computing provider to perform inference using a specified LLM, and the computing provider executes the inference with an LLM. Importantly, the computing provider and model provider are often distinct. For instance, the model provider could be a large tech company, such as Meta or DeepSeek, while the computing provider is typically an entity or individual with surplus computational resources.
>
> We have **revised the paper accordingly and encourage the reviewers to read the updated version for further clarity**.

---

> ### Author Response · Authors · 2024-11-21
> **Weakness 1: This weakness is due to a key misunderstanding of our setting.**
>
> **Weakness 1: It seems that model providers are already heavily disincentivized from “swapping models”...**
>
> **Re.:** We must emphasize that these questions and this weakness stem from a **fundamental misunderstanding** of our paper, where our **computing providers** are decentralized individuals and small companies who rent out their surplus computation capacity. **These are not model providers** like Meta or DeepSeek.
>
> As stated in the second paragraph of Section 1, we believe without oversight, such decentralized computing providers have a strong incentive to swap models because a smaller model demands significantly less memory and processing power, reducing costs substantially. Additionally, regardless of the model used, the providers ultimately return only a text output, making it challenging to detect any model swap. Given this, providers may be motivated to secretly swap the model, underscoring the necessity of a verification protocol to ensure compliance with the specified model.
>
> We acknowledge that our paper's motivation was not clearly presented, as this setting is entirely new. We have revised the paper to provide a clearer and more detailed explanation of the motivation. We welcome further discussion on this topic.

---

> ### Author Response · Authors · 2024-11-21
> **Weakness 2**
>
> **Weakness 2: For example, how are the inputs to the proxy task generated? Does the choice in the labeling function matter?**
>
> **Re.:** As shown in Figure 2 on page 4, the inputs to the proxy task are the model’s hidden representations, generated and provided by the computing provider.
>
> The choice of labeling function is indeed critical, as it directly affects the difficulty of the proxy task. If the task is too challenging, training may become impractical, impacting verification performance. Ideally, we aim to generate labels that the hidden representations should contain sufficient information to predict. In our secret-based protocol, we use a labeling network to produce the label vector, which enhances the property of secret distinguishability and removes the need for post hoc design.
>
> We have also revised Figure 2 to improve clarity and address any potential confusion.

---

> ### Author Response · Authors · 2024-11-21
> **Weakness 3**
>
> **Weakness 3: Lines 234-239 show that a trusted third party has to train several components, but the third party is only included in the advanced protocol and not the simple protocol.**
>
> **Re.:** Apologies for the confusion. In our original Figure 2, we only showed the role of the trusted third party **during protocol deployment**. The trusted third party indeed needs to train several components **before protocol deployment**, although this was not intended to be shown in Figure 2.
>
> To make it more accurate, we have revised Figure 2 to precisely reflect this, separating it into "before deployment" and ''during deployment''.

---

> ### Author Response · Authors · 2024-11-21
> **Weakness 4**
>
> **Weakness 4: And yet, what prevents an adversary from simply using a small model with the trusted third party to begin with?**
>
> **Re.:** We’re not certain we fully understand this question. Could you please clarify your concern? We would be happy to discuss this further.

---

> ### Author Response · Authors · 2024-11-21
> **Weakness 5**
>
> **Weakness 5: Limited security guarantees.**
>
> **Re.:** Rigorous security guarantees come with high computational costs, which are impractical for LLMs due to their inherent size. Our approach leverages a machine learning-based method to enhance computational efficiency while making only minimal sacrifices in security.
>
> To ensure the robustness of our proposed protocol, we discuss two strong adaptive attacks in Section 4.3, as well as many additional possible attacks in Appendix C. These adaptive attacks are strong ones that are carefully designed to fundamentally evaluate the security of our protocol. We also point out that we leverage a periodic secret update process as a defense mechanism—a commonly used technique in cryptography to enhance protocol safety, such as the rekeying process in WPA [1]. In Section 5.4, we empirically demonstrate that our protocol remains robust under proposed attacks, confirming its effectiveness.
>
> Additionally, we have revised Appendix A.1 to include a discussion of this limitation.
>
> [1] https://en.wikipedia.org/wiki/Rekeying_(cryptography)

---

> ### Author Response · Authors · 2024-11-21
> **Weakness 6**
>
> **Weakness 6: The defense requires releasing a (projection) of the model’s last-layer representation. This can make the model susceptible to model-stealing attacks.**
>
> **Re.:** This concern does not apply to our scenario, as we focus on **open-source** LLMs, as clearly stated in the title and introduction, aligning with the decentralized computing context we are considering. Since model parameters are publicly accessible, model-stealing is not an issue in this context.

---

> ### Author Response · Authors · 2024-11-21
> **Weakness 7**
>
> **Weakness 7: Have the authors considered the case where the provider only switches out some queries?**
>
> **Re.:** Thank you for raising this question. In the revised Appendix A.3, we provide a detailed discussion on addressing this scenario using a Bayesian framework combined with an Expectation-Maximization (EM) algorithm. The appendix includes comprehensive mathematical derivations that outline how to formulate the problem, infer latent states, and update parameters in this setting. We encourage the reviewers to refer to Appendix A.3 for the full explanation and derivations.

---

> ### Author Response · Authors · 2024-11-21
> **Question 1: This question is due to a key misunderstanding of our setting.**
>
> **Question 1: Many model providers are already training cheaper smaller models and showing how these models achieve near-par performance to larger models. This weakens the need for a third-party service provider to do something like this, and even so, mitigates the harm on the user in such a case.**
>
> **Re.:** This question appears to stem from a **fundamental misunderstanding** of the distinction between "computing provider" and "model provider." As discussed in the general response, these are distinct concepts: the model provider develops the LLM (e.g., a company like Meta), while the computing provider is an entity (often decentralized individuals or small companies) that rents out computational resources to users for running inference. This distinction is critical to understanding our setting.
>
> The development of smaller models with near-par performance does not eliminate the value of larger models. According to scaling laws, larger models generally yield better results when computational resources allow, and this performance gap remains non-negligible.
>
> When a user specifies and pays for a larger model, substituting it with a smaller one inherently results in a loss to the user. Even if the performance gap has narrowed, this does not negate the issue; it’s a fundamental business ethic that users should receive the service they pay for.

---

> ### Author Response · Authors · 2024-11-21
> **Question 2**
>
> **Question 2: Thus, the user can always test the performance of these models against their evals to determine if a bad model is being served. Thus, detection seems easy.**
>
> **Re.:** We have addressed this trivial solution in Appendix A.2, noting that it is highly vulnerable and easily bypassed.
>
> While the user could send a curated set of benchmark prompts to verify model performance, a malicious provider could circumvent this by detecting known benchmark prompts and selectively using the correct model only for those cases, while employing a different model for other queries. This approach also increases inference costs for the user since the evaluated examples are not what the user actually needs, making it an impractical solution.
>
> Furthermore, it is a fundamental business ethic that users should receive the service they pay for. Even if smaller models may achieve similar performance, this does not justify substituting them for the specified model. Users should have confidence that the model they requested is the one being used.

---

> ### Author Response · Authors · 2024-11-21
> **Question 3**
>
> **Question 3: The damages of being caught seem to far outweigh the benefits.**
>
> **Re.:** For decentralized computing providers, especially individuals or small companies, the damage from being exposed is likely far less severe than assumed. However, the potential benefit of model-swapping—significantly reduced computational costs—is substantial and cannot be ignored.
>
> Even if the claim holds, relying solely on the fear of being caught is overly optimistic. Practical incentives outweigh hypothetical risks. Financial audits, for example, remain essential despite severe penalties for misreporting, as the temptation to cut costs persists. Similarly, our protocol provides an essential layer of oversight, ensuring that users receive exactly the service they specify and pay for.

---

> ### Author Response · Authors · 2024-11-25
> **Rebuttal period ends soon – we anticipate your feedback!**
>
> Dear Reviewer jCVz,
>
> As the rebuttal/discussion period ends in a few days, we sincerely look forward to your feedback. We greatly appreciate the time and effort you have dedicated to reviewing our paper and helping us improve it.
>
> In the rebuttals, we have clarified **key misunderstandings** about the decentralized computing setting, addressed specific concerns regarding the roles of the trusted third party and computing providers, and provided empirical evidence to support the accuracy of our protocol. We have also revised the paper for better clarity.
>
> We would be grateful if you could review **our responses, additional results, and revised paper** to let us know whether they address or partially address your concerns. Your input on whether our explanations are on the right track would be invaluable.
>
> Please also let us know if there are any further questions or comments about this paper. We strive to consistently improve the paper and would deeply value your insights.
>
> Kind Regards,
>
> Authors of Submission 11602

---

> ### Author Response · Authors · 2024-12-01
> **Extended rebuttal period ends in 2 days – we anticipate your feedback!**
>
> Dear Reviewer jCVz,
>
> With the extended rebuttal/author discussion period ending in 2 days, we sincerely look forward to your feedback. We greatly appreciate the time and effort you have dedicated to reviewing our paper and helping us improve it.
>
> In the rebuttals, we have clarified **key misunderstandings** about the decentralized computing setting, addressed specific concerns regarding the roles of the trusted third party and computing providers, and provided empirical evidence to support the accuracy of our protocol. We have also revised the paper for better clarity.
>
> We would be grateful if you could review **our responses, additional results, and revised paper** to let us know whether they address or partially address your concerns. Your input on whether our explanations are on the right track would be invaluable.
>
> Please also let us know if there are any further questions or comments about this paper. We strive to consistently improve the paper and would deeply value your insights.
>
> Kind Regards,
>
> Authors of Submission 11602

---

> > ### Comment · Reviewer_jCVz · 2024-12-03
> > **Read rebuttal; maintaining score.**
> >
> > I have read the rebuttal and decide to maintain my score for the following reasons.
> >
> > **Weakness in motivation / threat model:** It is still unclear what the threat model is here and what the capabilities of each party are. It seems as though the individuals renting out compute are expected to provide an API whereby model providers/users query this. This seems highly unlikely that this would be the mode of operation. Instead, it is more likely that model providers provide a binary up -front to the decentralized workers. THough I can envision that such a SVIP scheme could be useful here, all of this detail is important and missing.
> >
> > **Agreement with Reviewer 84uz**: I agree with the proposed trivial solution and that there are simple baselines that should be explored. In addition to the proof-of-work literature, the proof-of-learning literature can also be looked at.
> >
> > Thank you for the other clarifications. These have cleared up some weaknesses and the paper has definitely improved in my opinion. Overall, I still believe that this paper needs a better motivation, threat model, and baselines.

---

> > > ### Author Response · Authors · 2024-12-03
> > > **Thank you for your feedback. We believe there are still key misunderstandings.**
> > >
> > > Thank you for your continued feedback. We appreciate your comments and believe there are still **key misunderstandings** regarding our setting and protocol.
> > >
> > > 1. **Capability of each party, and motivation**
> > >
> > > We want to again **describe our setting in the simplest language**:
> > >
> > > (1) *Computing providers* are decentralized entities, often small companies or individuals, renting out computational power at competitive prices.
> > >
> > > (2) Users seek to run expensive LLM inference with these decentralized providers at a low cost.
> > >
> > > (3) A trusted third party connects users and computing providers, ensuring that providers cannot cheat and that users receive the service they requested.
> > >
> > > *Model providers* (e.g., Meta, DeepSeek) are out of scope in our protocol.
> > >
> > > With the increasing popularity of LLMs, the demand for open-source LLM inference requests will rise, and the decentralized computing setting discussed in our paper offers a promising solution for low-cost inference. Real-world examples of such decentralized platforms already exist, as highlighted in our rebuttal.
> > >
> > > 2. **Unclear what the threat model is here**
> > >
> > > **Threat model in the simplest terms**: computing providers may not do the computing they are asked for, and swap larger models with smaller ones to save operational costs.
> > >
> > > Our paper discussed the setting from an API perspective for ease of understanding. However, it does not fundamentally differ from the binary upfront setting you mentioned, where the binary (e.g., a docker image) simply adds a layer of obscurity. In such a case, the computing provider could still inspect the binary to identify the specified LLM and use an alternative model to cheat. Thus, the binary upfront approach does not address this risk, underscoring the necessity of our verification protocol.
> > >
> > > 3. **Trivial solution proposed by Reviewer 84uz**
> > >
> > > Occasional checks by the third party, as suggested, result in **significantly higher error rates** compared to our protocol, and its cost is often impractical even if we only check a few percent of the queries. For example, if the computing provider cheats on every query and the third party examines only 10% of the queries, 90% of queries will escape detection. Moreover, the cost of our verification protocol, particularly in terms of GPU memory (1.4G) and inference time (0.0056 s), is significantly lower than the cost of performing even a single LLM inference, as demonstrated in Table 3.
> > >
> > > In our protocol, the third party requires only minimal and fixed computational power (as shown in Table 3) and is involved mainly **before deployment**. In contrast, the "trivial solution" imposes computational demands on the third party that scale with the number of queries and users. In practice, the third party cannot match the computational power of all decentralized providers, with potentially thousands or even more computers, making even 10% or 1% of occasional checks by the third party infeasible. The trivial solution simply **does not scale**.
> > >
> > > We have also discussed several baseline methods in Appendix A.2, demonstrating that many naive approaches are **highly vulnerable to attacks**. Since our work focuses on an adversarial setting with potentially malicious computing providers, our goal is to design a robust protocol, such as our proposed SVIP.
> > >
> > > We thank you for acknowledging the improvements in the paper and are open to further discussions to clarify any remaining concerns.

---

### Official Review · Reviewer_84uz · 2024-11-10

**Soundness:** 3
**Presentation:** 3
**Contribution:** 1
**Rating:** 3
**Confidence:** 4

**Summary:**

This paper studies verifiable inference for language models. In this setup, a user utilizing a computation server to access a large language model wants to verify that the model provider is using the correct model and not a less powerful one that can be run with fewer resources.

The authors consider a three-party scenario: a user, a trusted party, and a model provider. Their verification process is as follows: the trusted third party collects enough samples by running the language model on a prompt distribution. Then it stores all the intermediate outputs of the model. Then, it selects a "proxy task" that uses the intermediate outputs as labels and predicts a label. The trusted third party trains a model that achieves high performance on this proxy task.

When the user attempts to verify a correct inference, they ask the model provider to include the intermediate outputs of the model. These outputs are then sent to the trusted third party. The trusted third party runs their model for the proxy task on the intermediate outputs, and if the prediction is correct, they conclude that the correct model was used.

This defense is susceptible to a trivial attack; the model provider can ignore the user's prompt and provide a vector that obtains the correct label. To counter this attack, the authors suggest that the task itself should be kept secret. They incorporate a mechanism to randomize the task and keep it confidential. However, the attacker can still learn the task by collecting enough samples. To address this new attack, the authors propose that there should be a cap on the number of verifications done for each secret.

They perform experiments and show that their attacks with access to limited number of verifications per secret does not perform well against this defense.

**Strengths:**

- I find the topic of the paper interesting and important.

**Weaknesses:**

- The verification proposed in the paper is not useful in any reasonable scenario. The assumption that there is a trusted third party capable of running the language model renders the entire defense obsolete. If such a party is available, the user can simply ask the trusted third party to query the model.
- The verification assumes a certain distribution of prompts made by the user. Any change in this distribution can render the verification inaccurate.
- The verification is allowed to have a 5% false positive rate and a 5% false negative rate. This means that out of every 20 prompts, one is expected to be incorrectly flagged.
- More sophisticated attacks can be employed to recover the secret task.

**Questions:**

- I believe the verification setup of the paper is flawed. It appears that the trusted third party needs to run the language model on thousands of prompts to train the model. Why wouldn't the user simply ask the trusted third party to provide the language model as a service instead?

- Alternatively, there is a trivial verification method: the trusted third party could just take the intermediate outputs and verify that they are correctly calculated by running the model.

- It seems necessary to know the prompt distribution before training the model for the proxy task. How is this reasonable when using large language models (LLMs)? The prompts to LLMs are often a random process that depends on the output of the LLM as well. It would be challenging to approximate the distribution.

- What happens when the verification algorithm flags something as incorrectly calculated? Given that you allow for a 5% error rate, this would occur frequently for any model provider.

- I think the attacks you developed are not optimal. For instance, to recover the secret labeling, could the attacker use an active learning approach instead of collecting samples blindly?

---

> ### Author Response · Authors · 2024-11-21
> **Thank you for your feedback! We believe there are key misunderstandings in the review.**
>
> Thank you for your thoughtful feedback. In the rebuttals below, we have clarified **key misunderstandings** about the decentralized computing setting, addressed specific concerns regarding the roles of the trusted third party and computing providers, and provided empirical evidence to support the accuracy of our protocol.
>
> We have also **revised the paper accordingly and encourage the reviewers to read the updated version for further clarity**. We welcome further discussion on any remaining questions or suggestions.

---

> ### Author Response · Authors · 2024-11-21
> **Weakness 1 & Question 1: Clarification about our setting**
>
> **Weakness 1 & Question 1: The assumption that there is a trusted third party capable of running the language model renders the entire defense obsolete…Why wouldn't the user simply ask the trusted third party to provide the language model as a service instead?**
>
> **Re.:** We focus on the decentralized computing setting, where the third party can be understood as a platform connecting users and decentralized computing providers. Examples of such platforms include the Golem Network [1], Akash Network [2], Render Network [3], and Spheron Network [4]. In this setting, trust between users and computing providers is crucial. These platforms are motivated to monitor computing providers' compliance with specified model usage to maintain user trust. Without this oversight, users would likely lose confidence and leave the platform.
>
> It is important to emphasize that **the third party does not need significant computational power itself**. Its role is to facilitate and monitor the utilization of massive computational resources from **decentralized providers**. In our protocol, the third party incurs only manageable computational overhead, as shown in Table 3(b), and has minimal involvement **during the deployment stage** (Figure 2). The decentralized providers are primarily responsible for providing computational power during interactions.
>
> Lastly, if the third party directly provided computational power, it would simply act as a computing provider itself. This would not eliminate the need for external oversight, as the same risk of dishonest behavior—such as using a smaller model while charging for a larger one—would still exist.
>
> We have revised our paper to provide a clearer and more detailed motivation.
>
> [1] https://www.golem.network/
>
> [2] https://akash.network/
>
> [3] https://rendernetwork.com/
>
> [4] https://www.spheron.network/

---

> ### Author Response · Authors · 2024-11-21
> **Question 2**
>
> **Question 2: Alternatively, there is a trivial verification method: the trusted third party could just take the intermediate outputs and verify that they are correctly calculated by running the model.**
>
> **Re.:** While this approach is theoretically possible, it is impractical due to its excessive cost—it would essentially double the original computational cost for the user.
>
> Additionally, it would require the trusted third party to have computational resources comparable to the entire decentralized computing network, which contradicts the core purpose of decentralized computing and is infeasible in practice.

---

> ### Author Response · Authors · 2024-11-21
> **Question 3 & Weakness 2**
>
> **Question 3 & Weakness 2: It seems necessary to know the prompt distribution before training the model for the proxy task.**
>
> **Re.:** We have empirically demonstrated the robustness of our approach to shifts in prompt distribution. Specifically, in Section 5.2 and Appendix D.3, we show that the protocol trained on LMSYS-Chat-1M achieves low FPR and FNR on 2 additional datasets, namely ToxicChat and web questions, both of which exhibit distribution shifts. This evidence supports that our method does not require prior knowledge of the prompt distribution.

---

> ### Author Response · Authors · 2024-11-21
> **Question 4 & Weakness 3**
>
> **Question 4 & Weakness 3: The verification is allowed to have a 5% false positive rate and a 5% false negative rate.**
>
> **Re.:** As outlined in the general response above, the final conclusion about the provider’s honesty is not based on a single prompt but rather on **multiple distinct** prompt queries.
>
> Intuitively, even if an error occurs on one prompt (with a probability <5%), multiple independent prompt queries make it highly unlikely for errors to persist across the entire batch, leading to a highly reliable conclusion. As an analogy, **repeatedly** tossing a coin can provide a high-confidence bound on whether the coin is fair or not (or, in our case, whether the provider is using the specified model or not), even with randomness involved in each toss (a certain level of error in our method).
>
> By maintaining FPRs and FNRs below 5% for individual prompts, user can draw a **final conclusion** about the provider’s honesty with high confidence. In Appendix A.3, we present empirical results demonstrating that both the type-I and type-II error rates approach zero as the number of prompts increases. For instance, when the threshold is $\tau = 0.5$, the type-I and type-II error rates are $1.7 × 10^{-49}$ and 0.0, respectively.

---

> ### Author Response · Authors · 2024-11-21
> **Question 5 & Weakness 4**
>
> **Question 5 & Weakness 4: For instance, to recover the secret labeling, could the attacker use an active learning approach instead of collecting samples blindly?**
>
> **Re.:** Thank you for this suggestion. We agree with you that more sophisticated attacks are possible. As the first paper to propose verifiable LLM inference, our contribution focuses on introducing this new problem setting and establishing a strong baseline method that addresses several strong adaptive attacks. As an initial step in this area, we cannot cover all potential attack strategies and have to leave more advanced approaches, such as active learning-based attacks, for future work.

---

> ### Author Response · Authors · 2024-11-25
> **Rebuttal period ends soon – we anticipate your feedback!**
>
> Dear Reviewer 84uz,
>
> As the rebuttal/discussion period ends in a few days, we sincerely look forward to your feedback. We greatly appreciate the time and effort you have dedicated to reviewing our paper and helping us improve it.
>
> In the rebuttals, we have clarified **key misunderstandings** about the decentralized computing setting, addressed specific concerns regarding the roles of the trusted third party and computing providers, and provided empirical evidence to support the accuracy of our protocol. We have also revised the paper for better clarity.
>
> We would be grateful if you could review **our responses, additional results, and revised paper** to let us know whether they address or partially address your concerns. Your input on whether our explanations are on the right track would be invaluable.
>
> Please also let us know if there are any further questions or comments about this paper. We strive to consistently improve the paper and would deeply value your insights.
>
> Kind Regards,
>
> Authors of Submission 11602

---

> ### Author Response · Authors · 2024-12-01
> **Extended rebuttal period ends in 2 days – we anticipate your feedback!**
>
> Dear Reviewer 84uz,
>
> With the extended rebuttal/author discussion period ending in 2 days, we sincerely look forward to your feedback. We greatly appreciate the time and effort you have dedicated to reviewing our paper and helping us improve it.
>
> In the rebuttals, we have clarified **key misunderstandings** about the decentralized computing setting, addressed specific concerns regarding the roles of the trusted third party and computing providers, and provided empirical evidence to support the accuracy of our protocol. We have also revised the paper for better clarity.
>
> We would be grateful if you could review **our responses, additional results, and revised paper** to let us know whether they address or partially address your concerns. Your input on whether our explanations are on the right track would be invaluable.
>
> Please also let us know if there are any further questions or comments about this paper. We strive to consistently improve the paper and would deeply value your insights.
>
> Kind Regards,
>
> Authors of Submission 11602

---

> > ### Comment · Reviewer_84uz · 2024-12-02
> > **Thank you for the rebuttal**
> >
> > I appreciate the authors' effort in crafting a rebuttal. Unfortunately, I have to maintain my current score. I summarize the reasons behind my decision below:
> >
> > - Computational Power of the Trusted Third Party: The authors need to formalize the gap between the computational power of the third party and the first party. Specifically, the trivial solution I provided can be enhanced by only verifying certain steps of the computation at random, which makes the job of the verifier much less demanding. I encourage the authors to look at the literature on "Proof of Work," where ideas similar to this are explored.
> > - Trust Level in the Third Party: Part of the argument in the rebuttal is that the trusted third party cannot be relied upon to do the entire computation because they might cheat. If we are worried about the trusted third party cheating, then why aren't we worried about them cheating in the verification and verifying every query without doing the computation?
> > Weakness of Explored Attacks: I still believe the attacks explored are weak and need to be enhanced. Additionally, the robustness to distribution shift should be further explored. What does a 5% false positive rate mean in the distribution shift setting?
> > - Probabilistic Verification Concerns: I still don't understand how probabilistic verification can help identify bad actors. It seems users should be fine with approximately 5% of queries remaining unverified. Can the model provider now cheat by running the verifier themselves? Also, can they craft adversarial examples for the verifier?

---

> > > ### Author Response · Authors · 2024-12-03
> > > **Thank you for your feedback. We believe there are still fundamental misunderstandings.**
> > >
> > > Thank you for your detailed feedback. We believe there are still **fundamental misunderstandings** regarding our setting and protocol. Below, we address each point raised:
> > >
> > > 1. **Discussion on the trivial solution**
> > >
> > > Occasional checks by the third party, as suggested, result in **significantly higher error rates** compared to our protocol. For example, if the computing provider cheats on every query but the third party examines only 10% of the queries, 90% of queries will escape detection. Moreover, the cost of our verification protocol, particularly in terms of GPU memory (1.4G) and inference time (0.0056 s), is significantly lower than the cost of performing even a single LLM inference, as demonstrated in Table 3.
> > >
> > > In our protocol, the third party requires only minimal and **fixed** computational power (as shown in Table 3) and is involved mainly **before deployment**. In contrast, the "trivial solution" imposes computational demands on the third party that scale with the number of queries and users. In practice, the third party cannot match the computational power of all decentralized providers, making even 10% or 1% of occasional checks by the third-party infeasible. The trivial solution simply **does not scale**.
> > >
> > > 2. **Trust Level in the Third Party**
> > >
> > > In our previous response, we hypothesized that "if the third party directly provided computational power, it would simply act as a computing provider itself." This was intended to emphasize the essentiality of oversight, not to suggest mistrust in the third party.
> > >
> > > In practice, the platform (acting as the third party that connects the user and the computing provider) has a high incentive to maintain user trust to prevent abandonment of the service. Therefore, the assumption that the third party is trustworthy is valid within the scope of our decentralized computing setting.
> > >
> > > 3. **Weakness of Explored Attacks**
> > >
> > > We are the first to study this novel threat model, and we have listed many different types of attacks, including direct vector optimization attack, adapter attack, secret recovery attack, and fine-tuning attack. We also evaluated adaptive attacks (Section 5.4) and distribution shifts (Section 5.2). We believe the evaluation is quite comprehensive as the first work in this setting.
> > >
> > > 4. **Probabilistic Verification Concerns**
> > > - There is a misunderstanding here. The ultimate goal of users is to draw a **single final conclusion** about the provider’s honesty, rather than verifying individual queries. Let’s say we are a user and we found that our computing provider failed the verification on 900 out of 1000 queries we evaluated today. This intuitively gives us high confidence that the computing provider is dishonest, even if the outcome of each query has a certain degree of false positives or negatives. Another analogy is that you can fairly confidently decide a coin is unfair if you see 900 tops out of 1000 tosses, even if each toss gives you a random outcome. Formally, this is a hypothesis testing problem as we detailed in Appendix A.3.
> > > - There is no "verifier" in our protocol. Users themselves compare the labels and proxy task outputs to verify each query. While the computing provider could theoretically pretend to be a user and request a secret from the trusted third party, the likelihood of obtaining the same secret as the user is extremely low. Even if they manage this, crafting adversarial examples would be **prohibitively costly** and would outweigh the potential benefits of cheating.
> > > - Finally, as emphasized previously, there is no "model provider" in our setting. Instead, our focus is on the **computing provider**, which operates within a decentralized framework.
> > >
> > > We hope these clarifications address your concerns.

---

### Author Response · Authors · 2024-11-21
**General Response: We have clarified key misunderstandings about our setting, revised the paper and provided additional experiments.**

# General Response

We thank all the reviewers for their feedback. Since the setting discussed in our paper is quite novel and not commonly discussed in existing machine learning literature, and the concept of decentralized computing may sound new to many, we believe there are **clear misunderstandings** in the reviews. We have **revised our introduction section** based on the reviewer’s feedback to motivate our setting better and address the confusion. In this response, we would also like to briefly give a clear picture of our motivations.

## Motivation and setting
Our work is inspired by the recent trend of **decentralized computing**, where individuals and small companies with surplus computational resources can offer their services through computing rental platforms [1][2][3][4]. While this trend democratizes access to computational power, it also introduces trust challenges, particularly in ensuring that specified models are used as promised. Our proposed protocol addresses these challenges, making decentralized LLM inference both reliable and verifiable.

Our setting involves:
a user needing computational resources;
a computing provider (**not** the “model provider” or “LLM provider” as some reviewers may suggest), typically **decentralized** individuals or small companies with surplus computational capacity; they rent out computational power at competitive prices, but they do not develop LLMs themselves;
a trusted third party, typically the **rental platform** that connects users to decentralized computing providers. The third party does not need significant computational power itself - it aims to facilitate the utilization of massive computational resources from decentralized providers.

The user requests the computing provider to perform inference using a specified open-source LLM, and the computing provider executes the inference with an LLM. Importantly, the computing provider and model provider are often distinct. For instance, the model provider could be a large tech company, such as Meta or DeepSeek, while the computing provider is generally an individual entity with surplus computational resources.

## The assumption of a trusted third party
We emphasize that the assumption of a trusted third party is reasonable. The third party can be understood as a **platform** connecting computing providers and users, as seen with platforms such as the Golem Network [1], the Akash Network [2], the Render Network [3], and the Spheron Network [4]. These platforms are motivated to monitor computing providers' honesty during model inference to maintain user trust. Without such oversight, users would likely lose confidence and avoid using the platform.

Moreover, we also note that the computational overhead for the trusted third party is manageable, as shown in Table 3(b); for instance, even with the largest model size (70B), training the protocol requires less than 1.5 hours. This further supports the reasonableness of assuming a third party would be willing to provide the oversight and handle this overhead.

## Additional results of verification with multiple prompt queries
We emphasize that in practical scenarios, conclusions about a computing provider’s honesty are based on **multiple different** queries rather than a single one. In our protocol, a user can check the honesty of a computing provider in every query. A hypothesis testing framework can be adopted to combine the results of each individual query. With FPRs and FNRs below 5% for each individual query, the user can draw a **final conclusion** about the provider’s honesty with high confidence.

As an illustrative case, when using `Llama-3.1-70B` as the specified model and `Llama-2-7B` as the alternative, we achieve an FPR of 0.81% and an FNR of 3.13%. By **employing only 30 different queries, the type-I and type-II error rates of the final conclusion are effectively driven to near zero** across varying thresholds. For instance, when the threshold is $\tau = 0.5$, the type-I and type-II error rates are $1.7 × 10^{-49}$ and 0.0, respectively. This result demonstrates the strong robustness of our protocol. We direct the reviewers to Appendix A.3 for detailed explanations and results (Figure 5,6).

---

### Author Response · Authors · 2024-11-21
**(continued) General Response: We have clarified our setting, revised the paper and provided additional experiments.**

## Summary of paper revisions
We have revised our paper accordingly, with changes highlighted in blue. Specifically:
- The introduction section now includes a clearer motivation for our setting.
- Figures 1 and 2 have been updated to provide a more comprehensive visual illustration.
- Appendix A.3 has been expanded to discuss how hypothesis testing can be used to draw a high-confidence conclusion from multiple queries, supported by empirical results. Additionally, we introduced a Bayesian framework combined with an Expectation-Maximization (EM) algorithm to address scenarios where the computing provider occasionally switches to a smaller model.

Please note that our main technical contributions and experimental results remain unchanged.

[1] https://www.golem.network/

[2] https://akash.network/

[3] https://rendernetwork.com/

[4] https://www.spheron.network/

---

### Meta-Review · Area_Chair_qjmq · 2024-12-23

**Metareview:**

This paper introduces a verifiable inference protocol for LLMs. Different from prior work based on cryptographic techniques such as proof-of-work/learning, this protocol uses a secret proxy task that distinguishes final layer representations of the target model from other models.  This enables much more efficient verification of LLM computation compared to cryptographic solution, and allows the protocol to scale to production-grade models such as Llama-3.1-70B.

While reviewers appreciate the practicality of the protocol and the thorough evaluation performed in the paper, significant weaknesses remain even after the rebuttal:
1. The assumption of a trusted third party makes the protocol unrealistic for practical deployment.
2. Unclear protocol description and trust assumption.
3. Weak motivation for studying this problem.

AC believes these shortcomings severely undermine the paper's contribution and impact, and the paper is not ready for publication at this time. Nevertheless, the authors are encouraged to resubmit to a future venue after addressing the above shortcomings.

**Additional Comments On Reviewer Discussion:**

Reviewers appreciate the practicality of the protocol and the thorough evaluation performed in the paper, but significant weaknesses remain even after the rebuttal:
1. The assumption of a trusted third party makes the protocol unrealistic for practical deployment.
2. Unclear protocol description and trust assumption.
3. Weak motivation for studying this problem.

---

### Decision · Program_Chairs · 2025-01-22

Reject